# Microneutralization assay titer correlates analysis in two phase 3 trials of the CYD-TDV tetravalent dengue vaccine in Asia and Latin America

**Lindsay N. Carpp**[1]☉, **Youyi Fong**[1,2]☉, **Matthew Bonaparte**[3], **Zoe Moodie**[1], **Michal Juraska**[1], **Ying Huang**[1,2], **Brenda Price**[2], **Yingying Zhuang**[2], **Jason Shao**[1], **Lingyi Zheng**[3], **Laurent Chambonneau**[4], **Robert Small**[5], **Saranya Sridhar**[4], **Carlos A. DiazGranados**[6], **Peter B. Gilbert**[1,2]*

**1** Vaccine and Infectious Disease Division, Fred Hutchinson Cancer Research Center, Seattle, Washington, United States of America, **2** Department of Biostatistics, University of Washington, Seattle, Washington, United States of America, **3** Global Clinical Immunology, Sanofi Pasteur, Swiftwater, Pennsylvania, United States of America, **4** Clinical Sciences, Sanofi Pasteur, Marcy-L'Etoile, France, **5** Sanofi Pasteur, Orlando, Florida, United States of America, **6** Clinical Sciences, Sanofi Pasteur, Swiftwater, Pennsylvania, United States of America

☉ These authors contributed equally to this work.
* pgilbert@scharp.org

**Data Availability Statement:** Qualified researchers may request access to patient level data and related study documents including the clinical study

## Abstract

We previously showed that Month 13 50% plaque reduction neutralization test ($PRNT_{50}$) neutralizing antibody (nAb) titers against dengue virus (DENV) correlated with vaccine efficacy (VE) of CYD-TDV against symptomatic, virologically-confirmed dengue (VCD) in the CYD14 and CYD15 Phase 3 trials. While PRNT is the gold standard nAb assay, it is time-consuming and costly. We developed a next-generation high-throughput microneutralization (MN) assay and assessed its suitability for immune-correlates analyses and immuno-bridging applications. We analyzed MN and $PRNT_{50}$ titers measured at baseline and Month 13 in a randomly sampled immunogenicity subset, and at Month 13 in nearly all VCD cases through Month 25. For each serotype, MN and $PRNT_{50}$ titers showed high correlations, at both baseline and Month 13, with MN yielding a higher frequency of baseline-seronegatives. For both assays, Month 13 titer correlated inversely with VCD risk. Like $PRNT_{50}$, high Month 13 MN titers were associated with high VE, and estimated VE increased with average Month 13 MN titer. We also studied each assay as a valid surrogate endpoint based on the Prentice criteria, which supported each assay as a valid surrogate for DENV-1 but only partially valid for DENV-2, -3, and -4. In addition, we applied Super-Learner to assess how well demographic, Month 13 MN, and/or Month 13 $PRNT_{50}$ titers could predict Month 13–25 VCD outcome status; prediction was best when using demographic, MN, and $PRNT_{50}$ information. We conclude that Month 13 MN titer performs comparably to Month 13 $PRNT_{50}$ titer as a correlate of risk, correlate of vaccine efficacy, and surrogate endpoint. The MN assay could potentially be used to assess nAb titers in immunogenicity studies, immune-correlates studies, and immuno-bridging

report, study protocol with any amendments, blank case report form, statistical analysis plan, and dataset specifications. Patient level data will be anonymized and study documents will be redacted to protect the privacy of trial participants. Further details on Sanofi's data sharing criteria, eligible studies, and process for requesting access can be found at: https://www.clinicalstudydatarequest.com.

**Funding:** This study was supported by Sanofi Pasteur (https://www.sanofipasteur.com/). LNC, YF, ZM, MJ, YH, BP, YZ, JS, and PBG received a contract from Sanofi Pasteur to conduct the statistical analysis work and submit the results for publication. This study was also partially supported by the National Institute of Allergy and Infectious Diseases (https://www.niaid.nih.gov/) of the National Institutes of Health (award R37AI054165 to PBG). Sanofi Pasteur (http://www.sanofipasteur.com/en/) provided support in the form of salaries for authors (MB, LZ, LC, RS, SS, CADG) or indirectly as funding to the institutions to support the statistical analysis (LNC, YF, ZM, MJ, YH, BP, ZY, JS, PBG). Sanofi Pasteur also planned and performed the microneutralization assays and performed some of the data analysis. The specific contributions of each author are detailed in the "Author Contributions" section. Sanofi Pasteur reviewed and approved the final manuscript, but did not have any additional role in preparation of the manuscript.

**Competing interests:** The authors have read the journal's policy and have the following competing interests: MB, LZ, LC, RS, SS, and CDG are employees of Sanofi Pasteur. LNC, YF, ZM, MJ, YH, BP, YZ, JS, and PBG received a contract from Sanofi Pasteur to conduct the statistical analysis work and submit the results for publication. Sanofi Pasteur is the manufacturer of the CYD-TDV vaccine (Dengvaxia). This does not alter our adherence to PLOS ONE policies on sharing data and materials.

applications. Additional research would be needed for assessing the utility of MN titer in correlates analyses of other DENV endpoints and over longer follow-up periods.

## Introduction

Approximately 40% of the world is at risk of infection with the four serotypes of dengue virus (DENV-1, -2, -3, and -4) [1]. Symptomatic DENV infection can range in severity up to dengue hemorrhagic fever and dengue shock syndrome [2]. The global health and economic burdens of DENV are significant, with about 400 million (including 500,000 hospitalized) infections annually worldwide [1, 3] and an estimated annual $8.9 billion cost of dengue disease [4].

The CYD-TDV dengue vaccine (Dengvaxia®, Sanofi Pasteur) contains four recombinant, live attenuated chimeric viruses, each harboring the dengue premembrane/envelope genes of one serotype [5]. In two Phase 3 trials, CYD14 in 2–14-year-olds in Asia [6] and CYD15 in 9–16-year-olds in Latin America [7], CYD-TDV (or placebo) was administered at Months 0, 6, and 12. After the first injection, participants were followed-up with active surveillance for symptomatic, virologically confirmed dengue of any severity (VCD) until Month 25. Estimated vaccine efficacy (VE) of CYD-TDV against VCD caused by any serotype (DENV-Any) between Months 13 and 25 was 56.5% in CYD14 and 60.8% in CYD15 [6, 7], supporting licensing of CYD-TDV for individuals ≥9 years old in multiple dengue-endemic countries [8]. Subsequent analyses of VE by baseline dengue serostatus showed high estimated VE against hospitalized VCD and against severe VCD over 60 months in baseline-seropositive individuals; however, estimated VE against these two endpoints was negative in baseline-seronegative individuals (i.e., vaccinated baseline-seronegative individuals were at higher risk of these two endpints compared to unvaccinated baseline-seronegative individuals) [9]. The World Health Organization (WHO)'s Strategic Advisory Group of Experts on Immunization has concluded: "a 'pre-vaccination screening strategy' would be the preferred option, in which only dengue-seropositive persons are vaccinated" [10].

We recently conducted a case-cohort correlates analysis of 50% plaque reduction neutralization test ($PRNT_{50}$) neutralizing antibody (nAb) titers in CYD14 and CYD15 and showed that 1) high Month 13 $PRNT_{50}$ titers were associated with a lower rate of VCD between Months 13 and 25; and 2) estimated VE against VCD between Months 13 and 25 increased with Month 13 $PRNT_{50}$ titer [11]. However, estimated VE was positive for 9–16-year-old vaccine recipients with no or low Month 13 seroresponse and a few vaccine recipients with high seroresponse experienced breakthrough VCD, making $PRNT_{50}$ titers a "relative" correlate of protection [12].

The WHO has stated "Only the PRNT measures the biological parameter of in vitro virus neutralization. . .Newer tests measuring virus neutralization are being developed, but PRNT remains the laboratory standard against which these tests will need to be validated" [13]. However, the PRNT assay is low-throughput and difficult to automate. Alternative assays have been proposed [14–16], but whether and how nAb titers obtained by PRNT-alternative assays correlate with protection against VCD remain unknown. We developed and validated an enzyme-linked immunosorbent assay-based microneutralization (MN) assay. Compared to the PRNT assay, the MN assay requires less serum, is higher throughput, and uses an objective spectrophotometric readout. Here we: 1) assessed the correlation/concordance of MN and $PRNT_{50}$ assay readouts, at baseline and at Month 13; 2) assessed Month 13 MN nAb titers as correlates of risk (CoRs) of VCD in CYD14 and CYD15; 3) assessed Month 13 MN nAb titers as

correlates of VE (CoVEs) against VCD in CYD14 and CYD15; and 4) built models using baseline demographics, Month 13 $PRNT_{50}$ titers, and/or Month 13 MN titers to classify participants by VCD outcome status. Our approach for (2) and (3) mirrored that used for our previous correlates analysis of $PRNT_{50}$ titers in CYD14 and CYD15 [11].

## Materials and methods

### CYD14 and CYD15

In harmonized designs, healthy children and adolescents aged 2–14 (CYD14; ClinicalTrials. gov ID NCT01373281 [6]) or 9–16 (CYD15; ClinicalTrials.gov ID NCT01374516 [7]) were randomized (2:1) to vaccine or placebo, with randomization stratified by age group and site. Vaccinations were administered at Months 0, 6, and 12. Active surveillance for symptomatic VCD occurred from the day of the first injection to Month 25. [6, 7]. As in [11], correlates analyses were based on the primary study endpoint of symptomatic, virologically confirmed dengue of any serotype (DENV-Any) and on the serotype-specific DENV-1, -2, -3, and -4 VCD endpoints.

### Ethics statement

The trial protocols were approved by all relevant ethics review boards, and parents or guardians provided written informed consent and older children provided written informed assent before participation, in accordance with local regulations. All patient data were anonymized.

The ethics review boards for CYD14 were the following: The Committee of Medical Research Ethics, Faculty of Medicine, University of Indonesia, Jakarta, Indonesia; The Research and Development Unit Medical Faculty University of Udayana, Sanglah General Hospital, Denpasar, Indonesia; Health Research Ethics Committee, Faculty of Medicine University of Padjadjadrain, Dr Hasan Sadikin Hospital, Bandung, Indonesia; Medical Research and Ethics Committee, Ministry of Health, Malaysia, Kuala Lumpur, Malaysia; Research Institute for Tropical Medicine IRB, Alabang, Muntinlupa City, Philippines; Vicente Sotto Memorial Medical Center EC, Cebu City, Philippines; Chong Hua Hospital Institutional Review Board, Cebu City, Philippines; Walter Reed Army Institute of Research International Review Board (WRAIR IRB), MD, USA; The Ethical Review Committee for Research in Human Subjects, Ministry of Public Health, Thailand; Ethics Committee of the Faculty of Tropical Medicine, Mahidol University, Bangkok, Thailand; Pasteur Institute EC, Ho Chi Minh City, Vietnam.

The ethics review boards for CYD15 were the following: Comitê de Ética em Pesquisa do Centro Ciências da Saúde (CCS) da Universidade Federal do Espírito Santo (UFES) (CEP/CCS/UFES); Comitê de Ética em Pesquisa em Seres Humanos do Hospital Universitário Onofre Lopes / RN; Comitê de Ética em Pesquisa em Seres Humanos do Hospital das Clínicas da Universidade Federal de Goiás; Comitê de Ética em Pesquisa em Seres Humanos da Universidade Federal de Mato Grosso do Sul—UFMS; Comitê de Ética em Pesquisa em Seres Humanos da Universidade Federal do Ceará; Comissão Nacional de Ética Em Pesquisa—CONEP; Comité de Etica en la Investigación—CAIMED; Comité Corporativo de Ética en Investigación Fundación Santafe de Bogotá; Comité de Ética en Investigación Biomédica—CDI; Comité de Ética en Investigación Biomédica (CEIB) de la Unidad de Investigación Científica de la UNAH; Instituto Nacional de Pediatría Comité de Ética en Investigación; Instituto Nacional de Pediatría; Comité Ética y de Investigación—UV Universidad Veracruzana; Saluz Comité de Investigación y Bioética; Copernicus Group IRB—CGIRB.

WHO Universal Trial Numbers: U1111-1116-4957; U1111-1116-4986.

## Case-cohort sampling design and cohort definitions

Approximately 10% (CYD14) or 20% (CYD15) of all participants enrolled in the first 2 to 4 months of each trial were randomly assigned to the immunogenicity subset (IS) [described in [6, 7]]. Serum samples for assessing nAb responses were collected from participants in the IS on Months 0, 7, 13, and 25. The $PRNT_{50}$ assay was run on the stored Month 13 serum samples, after which samples were refrozen and thawed approximately 5 years later, at which time the MN assay was run. The sampling design for measurement of Month 13 MN titers is given in S1 Text. For all analyses that used MN ($PRNT_{50}$) data, participants in the IS who completed the active phase (Day 0 to Month 25) without experiencing the primary DENV-Any endpoint and for whom Month 13 MN ($PRNT_{50}$) data were available are defined as controls. As in [11], cases are defined as participants who experienced the DENV-Any endpoint between Month 13 and Month 25. Analyses were based on cases and controls with Month 13 MN ($PRNT_{50}$) data. As Month 0 serum samples were collected only for the IS, Month 0 MN titers could only be measured for 20.4% of all cases in CYD14 and 8.7% of all cases in CYD15.

## $PRNT_{50}$ assay

The $PRNT_{50}$ assay was performed using Vero cells (CCL-81 from the American Type Culture Collection, Manassas, VA; master and working banks of Vero cells were prepared in-house) as in [17]. $PRNT_{50}$ titer represents the highest dilution of serum at which $\geq$ 50% of dengue challenge virus in plaque counts was neutralized compared to the challenge virus-alone control wells, as determined by linear regression.

## MN assay

General comparisons of the PRNT and MN assays are provided in [18, 19]. In contrast to the $PRNT_{50}$ assay, the MN assay was performed in 96-well (vs 24-well) plates, the virus-serum inoculum was not removed after virus adsorption, liquid (vs semisolid) medium was applied after adsorption, and reduction in virus infectivity due to neutralization by antibodies present in serum samples was detected by successive addition and incubations of dengue serotype-specific monoclonal antibodies, anti-mouse Ig HRP conjugate, and a chromogenic substrate. The same serotype-specific anti-dengue monoclonal antibodies and virus strains were used in both assays.

Briefly, 2-fold serial dilutions of serum samples (starting at 1:5 dilution) were incubated with an equivalent volume of a constant challenge dose of virus (200 $TCID_{50}$ per well for each serotype) and incubated for 90 minutes at 37˚C. A separate virus titration plate was prepared to determine the 50% tissue-culture infective dose ($TCID_{50}$). After neutralization, the serum-virus mixture was added to pre-seeded Vero cell monolayers in 96-well plates and an additional 100 μl of cell culture medium was added without removal of the virus inoculum after adsorption. The plates were incubated at 37˚C for either 4–5 days (depending on the virus serotype). The target virus challenge dose and days of incubation post-infection were determined for each serotype in order to provide an optimal signal-to-noise ratio during the ELISA steps. After the incubation period was complete, the cell culture medium was removed from the plates. The cells were then fixed with 80% acetone and incubated at room temperature for 10–15 min, followed by blocking with 5% non-fat dry milk in PBS-Tween-20 wash buffer. Dengue serotype-specific monoclonal antibodies were added, followed by anti-mouse IgG HRP conjugate and TMB substrate. The reaction was stopped with 2N sulfuric acid and the optical density (OD) of each well at 450 nm (650 nm as the reference wavelength) was measured using a SpectraMax 384 microplate reader with SoftMax Pro software version 6.5.1.

The 50% neutralization titer of the test serum sample against each serotype was defined as the reciprocal of the test serum dilution for which the virus infectivity was reduced by 50% relative to the challenge virus dose (without any antibodies) introduced into the assay and was calculated using the formula: [(Average OD of Virus Control—Average OD of Cell Control)/2 + Average OD of Cell Control]. The MN titer for each test sample was interpolated by calculating the slope and intercept using the last dilution with an OD below the 50% neutralization point and the first dilution with an OD above the 50% neutralization point to determine the MN titer using the formula: [MN Titer = (50% neutralization point—intercept)/slope]. Neutralization titers are presented as continuous values. For both assays, the lower limit of quantitation was 10; values below this were set to 5. The average titer is the average of each participant's four serotype-specific $\log_{10}$ titers.

## nAb assay correlation and concordance

Rank correlations between MN and $PRNT_{50}$ titers were adjusted for age and country; correlations were calculated as in [20] using the PResiduals R package [21]. For analyzing concordance of the two assays with respect to baseline serostatus determination, Cohen's kappa was calculated.

## Immune correlates analyses

Month 13 MN titers were assessed as CoRs and CoVEs as in [11]. In brief, the CoR analyses were performed in each of the vaccine and placebo groups separately, relating VCD risk to a given Month 13 MN titer variable with a logistic regression model that accounted for the case-cohort sampling [22] and adjusted for age, sex, and country. Results are reported as odds ratios of DENV-Any, DENV-1, DENV-2, DENV-3, and DENV-4 VCD per $\log_{10}$ increase in Month 13 nAb titer. P values for testing DENV-1, DENV-2, DENV-3, and DENV-4 nAb titer as a CoR were adjusted across the 4 serotypes using family-wise error rate (Holm-Bonferroni [23]) and false-discovery rate (Q values [24]) adjustment, separately for each treatment group and each trial. All P values and Q values are 2-sided.

The CoVE analyses were performed using the VE curve-effect modification framework [25–27]. This framework assesses how VE changes over subgroups of vaccine recipients, where subgroups are defined by Month 13 nAb titers. The analyses used the Juraska et al. method [28], employed with hinge logistic regression models [29] when there was sufficient data, if not, linear logistic regression models were used. Advantages of the hinge models are summarized in reference [29]. VE curves were estimated with pointwise and simultaneous bootstrap-based Wald 95% confidence intervals (CIs).

## Super-Learner classification of DENV-Any outcome status

The Super-Learner algorithm, implemented with the SuperLearner R package [30], was used to construct best models of the conditional probability of DENV-Any occurrence by the Month 25 visit based on demographic features (age, sex, country-specific individual serotype rates) and Month 13 $PRNT_{50}$ and MN titer variables (DENV-1, -2, -3, and -4 titer readouts and the average, minimum, and maximum titers for each participant). Baseline demographics were included in all models. Only participants with complete Month 13 $PRNT_{50}$ and MN nAb titer data were included in the analysis (n = 2273; 212 cases in the vaccine group, 284 cases in the placebo group). Inverse probability of censored weighting [31] was employed to adjust for the case-cohort sampling and the restriction to 9–16-year-olds. Models of the conditional probability of DENV-Any occurrence by the Month 25 visit were built separately for the

vaccine and placebo groups using four input variable sets, aiming to maximize the cross-validated area under the receiver operating characteristic curve (CV-AUC) [32].

## Results

### Correlation/concordance of MN and PRNT$_{50}$ titers

**High correlation between MN and PRNT$_{50}$ titers.**   Fig 1 shows scatterplots of PRNT$_{50}$ and MN titers from CYD14 (2–14-year-old) or CYD15 (9–16-year-old) participants at baseline (Month 0) and at Month 13 (1 month post-final dose). For serotype-specific nAb titer pairs, Spearman correlation coefficients were high (0.83–0.95), across all serotypes, trials, and time-points. Within each trial, serotype-specific correlations tended to be lowest for DENV-4, at both time-points. For each trial and each time-point, correlations were highest between average MN and average PRNT$_{50}$ titer.

For 9–16-year-olds in CYD14 and CYD15, covariate-adjusted Spearman rank correlations between PRNT$_{50}$ and MN titers were consistently high at both baseline and Month 13, and across the vaccine and placebo groups at Month 13 (S1 Table). Correlations were highest for the average titer at both time-points (baseline 0.96, 95% CI 0.95–0.97; Month 13 (placebo) 0.95, 95% CI 0.93–0.96; Month 13 (vaccine) 0.94, 95% CI 0.93–0.95), with high correlations for the DENV-1, -2, and -3 readouts (0.88–0.94 across both time-points). Correlations were slightly lower for DENV-4 (baseline 0.84, 95% CI 0.81–0.88; Month 13 (placebo) 0.87, 95% CI 0.84–0.90; Month 13 correlation (vaccine) 0.81, 95% CI 0.78–0.83).

**Substantial agreement in baseline serostatus classification between the two assays, but a higher proportion of individuals test baseline dengue-seronegative by the MN assay.**   It is

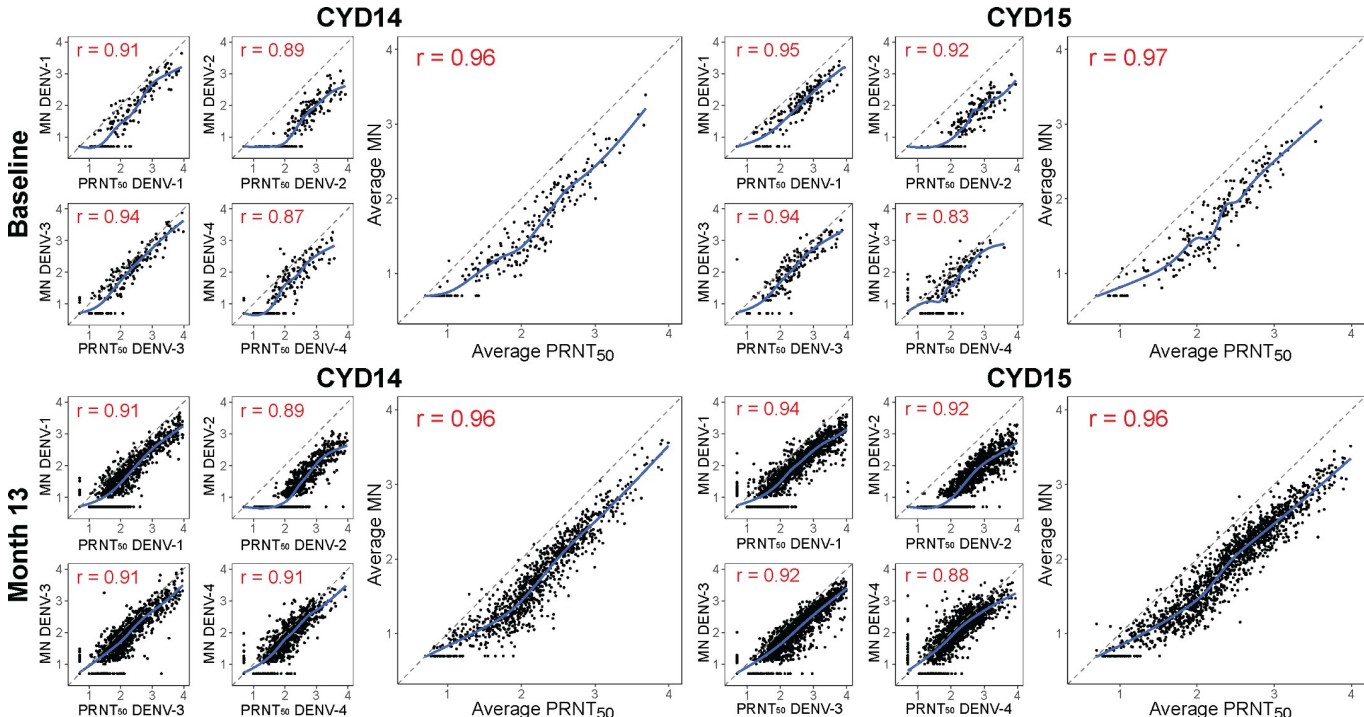

**Fig 1. Correlation of PRNT and MN nAb titers.** Pairwise plots of PRNT and MN nAb titers in each trial are shown for each serotype and for the average (geometric mean) across all four serotypes. Top row, baseline titers; bottom row, Month 13 titers. Spearman correlation coefficients are shown in red in the upper left of each panel. Plots display values from all participants in CYD14 (2–14-year-olds) and CYD15 (9–16-year-olds) for whom both MN and PRNT$_{50}$ nAb titers were available. The blue line in each plot is a loess fit to the points.

of interest to present vaccine safety and vaccine efficacy results stratified by baseline dengue serostatus [10, 33]. We examined the concordance of the two assays with respect to baseline serostatus classification and found good agreement (Cohen's κ = 0.81 across both trials, S2 Table, part A). Similarly agreement was found in each study (Cohen's κ = 0.80, CYD14; Cohen's κ = 0.82, CYD15; S2 Table, parts B,C) and across age groups (all Cohen's κ>0.8; S2 Table, parts D-F). The disagreement between the two assays was consistent in that the proportion of MN-/$PRNT_{50}$+ participants always exceeded the proportion of MN+/$PRNT_{50}$- participants (S2 Table). Potential explanations are increased specificity of the MN assay compared to the $PRNT_{50}$ assay, decreased sensitivity of the MN assay compared to the $PRNT_{50}$ assay, or both.

For 9–16-year-olds across both trials, the percentage of participants testing baseline-seronegative by the MN assay was higher for average titer and for each of the 4 serotype-specific titers than that testing baseline-seronegative by the $PRNT_{50}$ assay (all P<0.01, McNemar's test; S1 Fig, panel A). This difference was greatest for DENV-2 (43% vs 27%). This pattern continued for the placebo group when Month 13 titers were assayed, i.e. higher percentages of participants tested Month 13-seronegative by the MN assay compared to the $PRNT_{50}$ assay, across all five titer measurements (all P<0.01; S1 Fig, panel B). Similar results were obtained in the vaccine group for Month 13 DENV-1 and DENV-2 titers in that significantly greater percentages of participants tested Month 13-seronegative for DENV-1 and for DENV-2 by the MN assay compared to the $PRNT_{50}$ assay (P<0.01 for both; S1 Fig, panel C). However, no significant difference in Month 13-seronegativity rates between the two assays was seen in the vaccine group for average titer, DENV-3, or DENV-4 (P>0.05 for all), with seronegativity rates ≤ 2% across the two assays (S1 Fig, panel C).

## Correlates analysis using MN nAb measurements

**Case-cohort sampling scheme.** Using the approach described in [11], we analyzed Month 13 MN titer as a CoR of VCD and as a CoVE against VCD in CYD14 and CYD15. The case-cohort sampling design is shown in Fig 2; further details are given in S1 Text. The cohort for inference consisted of all participants who had not experienced VCD due to any DENV serotype (DENV-Any) by Month 13, who were randomly sampled into the IS, and who had an available MN titer measurement.

**Month 13 MN titer and Month 13 $PRNT_{50}$ titer perform comparably as CoRs of VCD.** We previously reported that Month 13 $PRNT_{50}$ titers were inverse CoRs of VCD in each trial and in each treatment group, as assessed using Cox proportional-hazards and logistic-regression models [11]. Here we used logistic regression models to determine the estimated odds ratios (ORs) of matched-serotype VCD in each trial per $\log_{10}$ increase in Month 13 nAb titer, adjusting for sex and country, for both assays. In both CYD14 and CYD15, average MN titer was a significant CoR for VCD of any serotype (both P<0.001). Serotype-specific MN titers were also significant CoRs for the serotype-matched VCD endpoints across both trials, except for DENV-3 in CYD14 (P = 0.068) and DENV-4 in CYD14 (P = 0.125) (Table 1). The performance of MN vs $PRNT_{50}$ titer as a CoR differed somewhat between the two trials, with MN titer a consistently stronger CoR than $PRNT_{50}$ titer across serotypes in CYD15, but with $PRNT_{50}$ titer outperforming MN titer for DENV-3 and DENV-4 in CYD14. The Month 13 MN and $PRNT_{50}$ nAb titer distributions of participants in the placebo and vaccine groups of CYD14 and CYD15, stratified by case (matched-serotype/non-matched serotype) control status, are shown in Figs 3 and 4, respectively.

**Month 13 nAb titer association with VCD is weak or absent for low nAb titers.** We previously reported that while high $PRNT_{50}$ titers were associated with high VE against VCD,

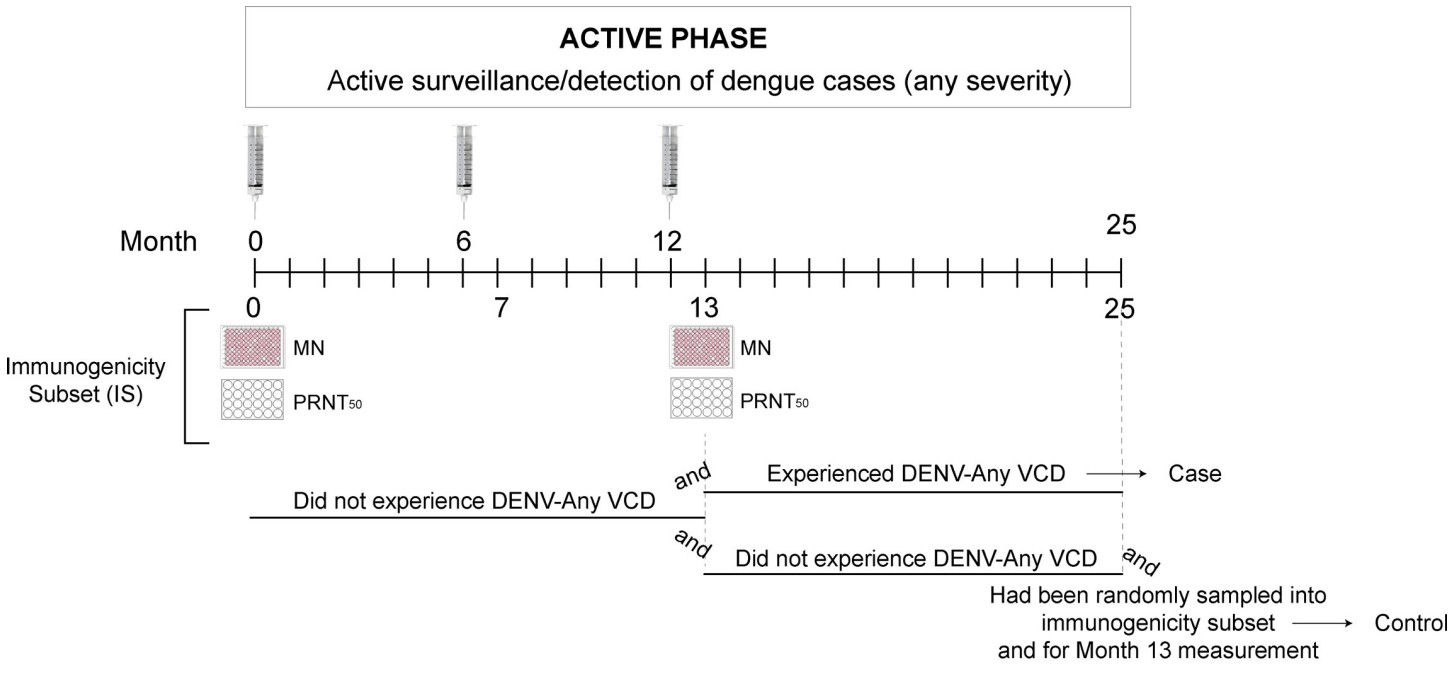

Fig 2. Case-cohort sampling scheme. Controls are defined as participants who were randomly sampled into the immunogenicity subset (IS), had an available titer measurement, and completed the active phase (25 months post-first vaccination) without experiencing the DENV-Any endpoint. For CYD14, the IS consisted of a random sample of participants enrolled in the first 2 months of the trial (randomized 2:1 for inclusion), corresponding to ~20% of the total CYD14 participants; for CYD15, the IS consisted of a random sample of participants enrolled in the first 2 to 4 months of CYD15 (randomized 1:1 for inclusion), corresponding to ~10% of the total CYD15 participants. The sampling design for measurement of Month 13 MN titers in CYD14 and CYD15 participants is detailed in S1 Text. As in Moodie et al. [11], cases are defined as participants who experienced the DENV-Any endpoint between Month 13 and Month 25. The table on the bottom shows the numbers of participants with nAb titer data available at Month 0 and at Month 13 in each trial, by case-control status.

for all serotypes, age groups, and across both trials, estimated VE against DENV-Any VCD was about 35% in 9–16-year-old vaccinees with no Month 13 seroresponse [11]. This finding suggested that other immune responses may be important for protection when nAb titers are low and that the association between VCD risk and nAb titer may be weaker (or absent) at low titers. We used generalized additive models with smoothing splines to assess how the estimated log odds of DENV-Any varied in 9–16-year-olds according to Month 13 MN average nAb titer, separately in the vaccine and placebo groups pooled across both trials. The results showed that DENV-Any risk was not logit linear with MN titer (P<0.001 in the vaccine and placebo groups, unpublished data), suggesting that titers below a certain threshold were not (or only weakly) associated with VCD risk. As hinge models fit the data more adequately than logit

**Table 1. Univariate logistic regression odds ratios of DENV-Any, DENV-1, DENV-2, DENV-3, and DENV-4 VCD in the vaccine groups of the CYD14 and CYD15 studies per $log_{10}$ increase in Month 13 nAb titer.**

| Assay | CYD14 (n = 1390 Vaccine Recipients) | | | | CYD15 (n = 1458 Vaccine Recipients) | | | |
|---|---|---|---|---|---|---|---|---|
| | **Titer variable, DENV endpoint** | | | | | | | |
| **Assay** | **Odds Ratio (95% CI)** | **P-value** | **Holm Adj. P-value** | **Q-value** | **Odds Ratio (95% CI)** | **P-value** | **Holm Adj. P-value** | **Q-value** |
| | **Average titer[a], DENV-Any** | | | | | | | |
| $PRNT_{50}$ | 0.24 (0.16, 0.37) | <0.001 | <0.001 | <0.001 | 0.15 (0.11, 0.21) | <0.001 | <0.001 | <0.001 |
| MN | 0.21 (0.13, 0.33) | <0.001 | <0.001 | <0.001 | 0.14 (0.10, 0.20) | <0.001 | <0.001 | <0.001 |
| | **DENV-1 titer, DENV-1** | | | | | | | |
| $PRNT_{50}$ | 0.39 (0.26, 0.58) | <0.001 | <0.001 | <0.001 | 0.31 (0.23, 0.42) | <0.001 | <0.001 | <0.001 |
| MN | 0.08 (0.03, 0.21) | <0.001 | <0.001 | <0.001 | 0.23 (0.15, 0.33) | <0.001 | <0.001 | <0.001 |
| | **DENV-2 titer, DENV-2** | | | | | | | |
| $PRNT_{50}$ | 0.42 (0.23, 0.76) | 0.004 | 0.006 | 0.004 | 0.18 (0.12, 0.27) | <0.001 | <0.001 | <0.001 |
| MN | 0.23 (0.13, 0.43) | <0.001 | <0.001 | <0.001 | 0.14 (0.08, 0.26) | <0.001 | <0.001 | <0.001 |
| | **DENV-3 titer, DENV-3** | | | | | | | |
| $PRNT_{50}$ | 0.38 (0.22, 0.63) | <0.001 | <0.001 | <0.001 | 0.41 (0.27, 0.64) | <0.001 | <0.001 | <0.001 |
| MN | 0.55 (0.29, 1.04) | 0.068 | 0.136 | 0.091 | 0.23 (0.13, 0.41) | <0.001 | <0.001 | <0.001 |
| | **DENV-4 titer, DENV-4** | | | | | | | |
| $PRNT_{50}$ | 0.31 (0.14, 0.67) | 0.003 | 0.006 | 0.004 | 0.22 (0.09, 0.51) | <0.001 | <0.001 | <0.001 |
| MN | 0.42 (0.14, 1.27) | 0.125 | 0.136 | 0.125 | 0.08 (0.03, 0.19) | <0.001 | <0.001 | <0.001 |

Models were adjusted for age, sex, and country.

[a]Geometric mean of the antibody titers against DENV-1, DENV-2, DENV-3, and DENV-4.

linear models for most analyses, hinge models were considered for estimating VE when there was sufficient data, if not, logit linear models were employed.

**Similar DENV-Any VE curves by Month 13 MN titer and by Month 13 $PRNT_{50}$ titer, except possibly at mid-range titers in both trials and at very low titers in CYD15.** Using the Juraska et al. method [28], vaccine recipients in CYD14 had an apparently stable level of VE against DENV-Any of approximately 20% for Month 13 average MN titers ranging from below the lower limit of detection to the hinge point at 21, after which VE increased as Month 13 average MN titer increased, reaching near 95% for the highest titers (Fig 5A). This pattern was similar to that previously observed for VE by Month 13 average $PRNT_{50}$ titer [Fig 5A in [11]]. Fig 5B plots the difference in VE against DENV-Any by MN titer minus VE against DENV-Any by $PRNT_{50}$ titer in CYD14, where the only difference is observed for titers around 50–80. As the MN Month 13 average titer distribution is shifted leftward compared to the $PRNT_{50}$ Month 13 average titer distribution, we conjecture that in this mid-range a lower value of MN captures the same information about VE as a higher value of $PRNT_{50}$. In CYD15, potentially negative to zero VE against DENV-Any was observed at Month 13 average MN titers ranging from below the lower limit of detection to around 10, after which VE leveled off to around 25% until the hinge point at 24. For Month 13 average MN titers greater than 24, VE increased as Month 13 average MN titer increased, reaching near 100% for the highest titers (Fig 5C). The VE-by-Month 13 average $PRNT_{50}$ curve from our previous analysis [Fig 5B in [11]] looked similar for titers greater than the hinge point, but in contrast to the MN VE curve, VE appeared stable around 25% for Month 13 $PRNT_{50}$ titers ranging from below the lower limit of detection to the hinge point at 61. Fig 5D plots the difference in VE against DENV-Any by MN titer minus VE against DENV-Any by $PRNT_{50}$ titer in CYD15. While the same difference observed in CYD14 is seen for titers around 50–80, in CYD15 an additional and larger difference in VE is seen for titers below the limit of detection. The magnitude of this

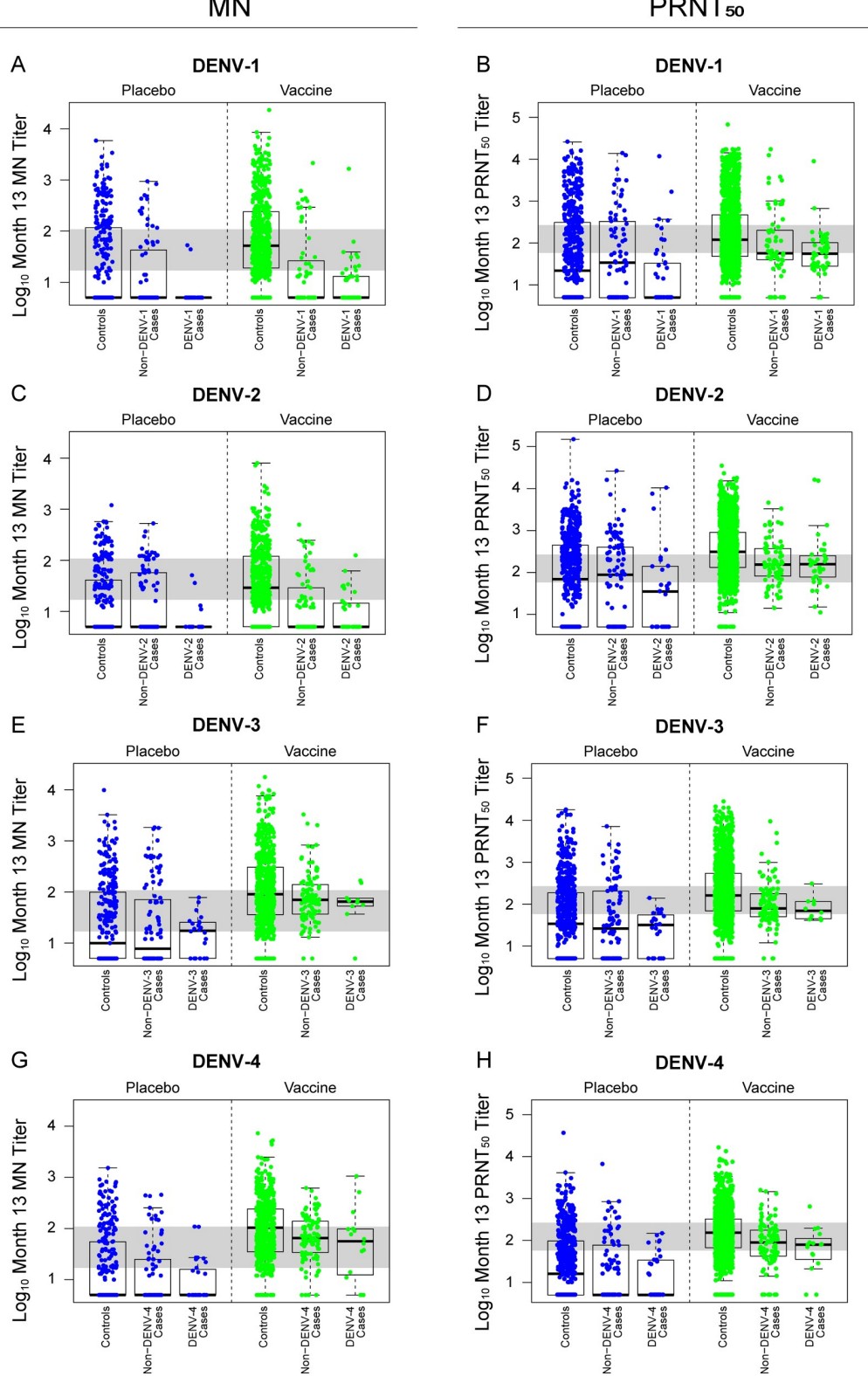

**Fig 3. Distributions of log$_{10}$ Month 13 MN (Panels A, C, E, G) and PRNT$_{50}$ (Panels B, D, F, H) titers in CYD14 participants (all ages = 2 to 14 years old), stratified by treatment group and case status (serotype-matched vs serotype-mismatched cases).** The gray horizontal shaded band denotes the middle third of nAb responses (log$_{10}$ MN titer = 1.23 − 2.03; log$_{10}$ PRNT$_{50}$ titer 1.76 − 2.42). A "matched-serotype case" is one where the VCD-causing virus was of the same serotype as the virus used in the nAb assay; a "nonmatched-serotype case" is one where the VCD-causing virus was of a different serotype than the virus used in the nAb assay.

difference decreases as the titers approach the limit of detection, after which the two curves appear similar. It is possible that the MN assay may be better at detecting a lack of VE in base-line seronegative individuals than the PRNT$_{50}$ assay. Alternatively, in CYD15, the VE curve estimation for vaccine recipients with low to no seroresponse could be unstable due to sparse data.

**nAbs measured by the MN assay may be better at mediating VE against DENV-Any than nAbs measured by the PRNT$_{50}$ assay.** We assessed VE against DENV-Any and against each of the serotype-specific endpoints for vaccine recipients with no Month 13 seroresponse (defined as titer<10 for all serotypes for DENV-Any; DENV-1 titer<10 for DENV-1 VCD, etc.), separately in each trial. For CYD14, all point and interval estimates of VE (except potentially those for VE against DENV-4 VCD) are consistent with full mediation with nAbs measured by the PRNT$_{50}$ assay (Table 2, part A). However, the VE point estimates against DENV-3 and DENV-4 VCD for individuals without MN-measured DENV-3 or DENV-4 serore-sponse, respectively, were higher than their PRNT$_{50}$ counterparts—suggesting that, for these two serotypes, nAbs measured by the MN assay may not fully mediate VE against their respec-tive endpoints. For CYD15, the lower bound of the 95% CI for VE>0% was above 0 for DENV-3 (MN) and for DENV-3 and DENV-4 (PRNT$_{50}$) (Table 2, part B). The DENV-4 result suggests that the MN assay may be identifying the lack of VE better than the PRNT$_{50}$ assay. For 9–16-year-olds pooled across both trials, nAbs measured by the MN assay may be better at mediating VE against DENV-Any than those measured by the PRNT$_{50}$ assay (Table 2, part C). However, the evidence for positive VE against DENV-3 and DENV-4 VCD in MN-DENV-3 and MN-DENV-4 Month 13 seronegative vaccinees again suggests that nAbs measured by the MN assay do not fully mediate VE against the DENV-3 and DENV-4 endpoints.

We also applied the Prentice criteria [34] to evaluate whether (or how closely) each Month 13 serotype-specific nAb response satisfied the Prentice definition of a valid surrogate end-point for the matched-serotype VCD outcome, in CYD14 and CYD15 together. Two Prentice criteria are readily supported across the nAb titer markers (serotype-specific VE > 0% and the marker correlates with VCD in each treatment group; S3 Table columns 2 and 3). The key third Prentice criterion is that treatment group does not predict VCD after accounting for the marker and adjusting for baseline variables that predict both the marker and VCD. Fig 6 shows the logistic regression estimates of cumulative endpoint rates for serotype-specific VCD and sampling weighted distributions of serotype-specific log$_{10}$ nAb titers, in CYD14 and CYD15 together, separately by Month 13 serotype-specific PRNT$_{50}$ titer and by Month 13 serotype-specific MN titer. The modeling results were consistent across both assays for all 4 serotypes, with results supporting (1) DENV-1 titer adheres remarkably well to the Prentice criteria (e.g., overlapped vaccine and placebo curves in panels A and B in Fig 6), (2) DENV-3 titer has a similar inverse association with VCD in each treatment group but departs from the third criterion with titer and treatment jointly predicting VCD; and (3) the DENV-2 and DENV-4 CoRs were significantly modified by treatment group, indicating departure from the third criterion. Regarding point (3), the cumulative endpoint rates of DENV-2 VCD by Month 13 DENV-2 PRNT$_{50}$ titer (Panel C of Fig 6) suggest that CYD-TDV vaccination could have increased DENV-2 VCD risk at lowest Month 13 DENV-2 PRNT$_{50}$ titers. Moodie et al. [11]

# CYD15 (9 to 16-year-olds)

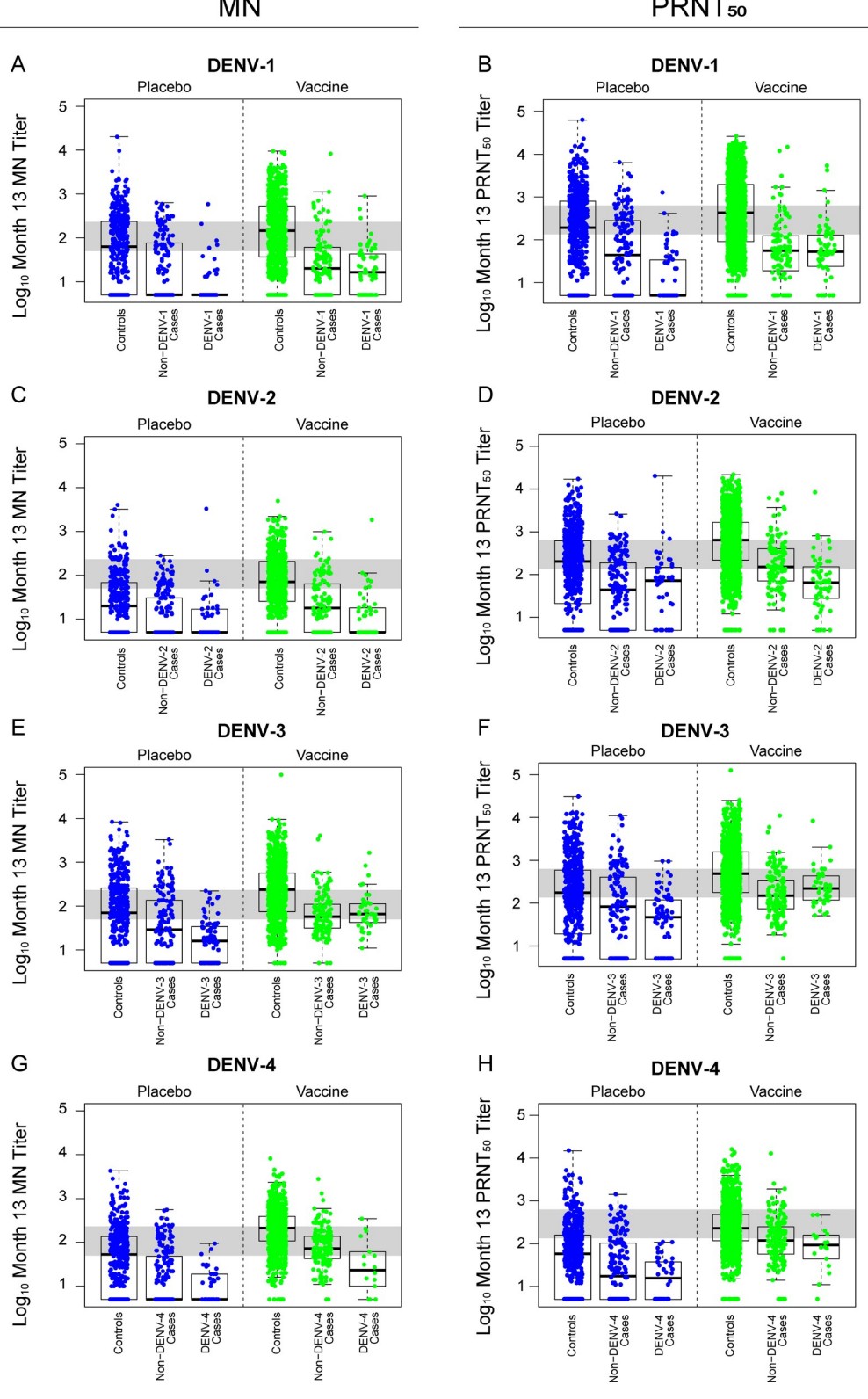

Fig 4. Distributions of $\log_{10}$ Month 13 MN (Panels A, C, E, G) and $PRNT_{50}$ (Panels B, D, F, H) titers in CYD15 participants (all ages = 9 to 16 years old), stratified by treatment group and case status (serotype-matched vs serotype-mismatched cases). The gray horizontal shaded band denotes the middle third of nAb responses ($\log_{10}$ MN titer = 1.7 − 2.36; $\log_{10}$ $PRNT_{50}$ titer = 2.13 − 2.8). A "matched-serotype case" is one where the VCD-causing virus was of the same serotype as the virus used in the nAb assay; a "nonmatched-serotype case" is one where the VCD-causing virus was of a different serotype than the virus used in the nAb assay.

previously addressed this issue, noting that simultaneous 95% confidence bands for DENV-2 VE include 0%, and an inference of vaccine-increased risk is based on a very small number of vaccine-recipient DENV-2 cases (0 DENV-2 cases among the 87 vaccine-recipients with no Month 13 $PRNT_{50}$ DENV-2 seroresponse in CYD14 and 5 DENV-2 cases among the 264 vaccine-recipients with no Month 13 $PRNT_{50}$ DENV-2 seroresponse in CYD15, with "no Month

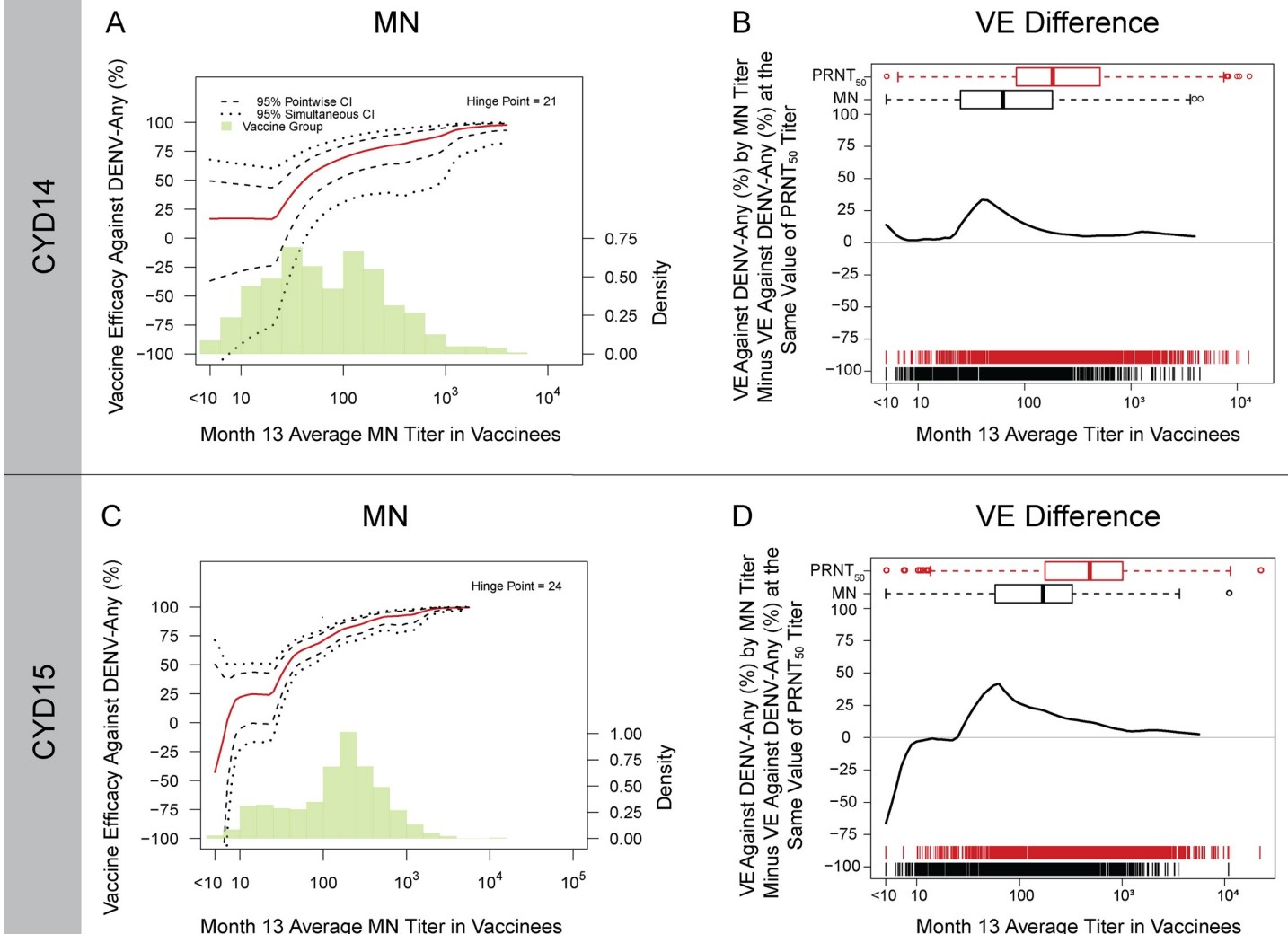

Fig 5. Estimated vaccine efficacy (VE) against DENV-Any by Month 13 MN titer in the vaccine group in A) CYD14 and C) CYD15. 95% pointwise and 95% simultaneous confidence intervals are also shown. Panels B (CYD14) and D (CYD15) show the estimated VE against DENV-Any by Month 13 MN titer minus estimated VE against DENV-Any at the same value of Month 13 $PRNT_{50}$ titer. [The VE curves by Month 13 $PRNT_{50}$ titer by themselves are shown in Moodie et al. [11].] DENV-Any = symptomatic, virologically confirmed dengue of any serotype, occurring between Month 13 and Month 25.

**Table 2. Comparison of point and interval estimates of VE as assessed by the MN vs. PRNT$_{50}$ assay in vaccine recipients with no Month 13 seroresponse.**

**A. CYD14**

| Endpoint | MN | | PRNT$_{50}$ | |
|---|---|---|---|---|
| | $\widehat{VE}$ (no Month 13 seroresponse*) (%) | 95% CI | $\widehat{VE}$ (no Month 13 seroresponse*) (%) | 95% CI |
| DENV-Any | 17 | (-38, 49) | 2 | (-49, 36) |
| DENV-1 | -122 | (-743, 42) | 5 | (-69, 46) |
| DENV-2 | -6 | (-164, 57) | -14 | (-159, 50) |
| DENV-3 | 66 | (-7, 89) | -39 | (-235, 42) |
| DENV-4 | 54 | (-82, 88) | 35 | (-88, 77) |

**B. CYD15**

| Endpoint | MN | | PRNT$_{50}$ | |
|---|---|---|---|---|
| | $\widehat{VE}$ (no Month 13 seroresponse*) (%) | 95% CI | $\widehat{VE}$ (no Month 13 seroresponse*) (%) | 95% CI |
| DENV-Any | -43 | (-311, 50) | 23 | (-5, 43) |
| DENV-1 | 12 | (-37, 44) | 27 | (-12, 52) |
| DENV-2 | -52 | (-149, 7) | -84 | (-216, -7) |
| DENV-3 | 59 | (31, 76) | 64 | (35, 81) |
| DENV-4 | 34 | (-146, 82) | 74 | (46, 87) |

**C. CYD14 and CYD15 9–16-year-olds**

| Endpoint | MN | | PRNT$_{50}$ | |
|---|---|---|---|---|
| | $\widehat{VE}$ (no Month 13 seroresponse*) (%) | 95% CI | $\widehat{VE}$ (no Month 13 seroresponse*) (%) | 95% CI |
| DENV-Any | 19 | (-23, 47) | 35 | (7, 54) |
| DENV-1 | 15 | (-27, 43) | 23 | (-9, 45) |
| DENV-2 | -31 | (-110, 18) | -47 | (-147, 13) |
| DENV-3 | 62 | (30, 79) | 56 | (28, 73) |
| DENV-4 | 62 | (0, 86) | 76 | (59, 85) |

* No Month 13 seroresponse = Month 13 titer below the lower limit of quantitation, set to 5.

13 PRNT$_{50}$ DENV-2 seroresponse" defined as Month 13 PRNT$_{50}$ DENV-2 titer < 10). Panel D of Fig 6 shows less concern for potential vaccine-increased DENV-2 VCD risk for individuals with lowest DENV-2 titers based on the MN assay, given that the vaccine and placebo curves are more similar in the left-tail of the plot. For DENV-4, cumulative endpoint rates decreased with increasing Month 13 serotype-matched titers in both treatment groups, with low cumulative DENV-4 rates at low titers (Panels G and H of Fig 6), and a borderline significant result that the rate was lower in the vaccine group at low titers. Together, these results show that Month 13 PRNT$_{50}$ titer and Month 13 MN titer are consistent with the Prentice criteria for DENV-1 but not for the other serotypes. The other evaluation statistics in S3 Table, and a comparison of the curves in Fig 6, support imperfect but substantial partial surrogate value for DENV-3 and DENV-4, and less so for DENV-2.

## VE against DENV-Any is positive and increases with average Month 13 MN titer, in both baseline-seropositive and baseline-seronegative subgroups

We previously showed that estimated VE against DENV-Any was approximately 25% for vaccine recipients with no seroresponse (defined as PRNT$_{50}$ titer less than the assay lower limit of quantification, 10, for all four serotypes) at Month 13 and increased similarly with average Month 13 PRNT$_{50}$ titer in baseline seronegative vs. baseline seropositive subgroups in CYD14 and CYD15 9–16-year-old vaccine recipients (baseline serostatus determined by the PRNT$_{50}$ assay) (Fig 7A and 7B; reproduced with modification from [11]). Using the same method [35],

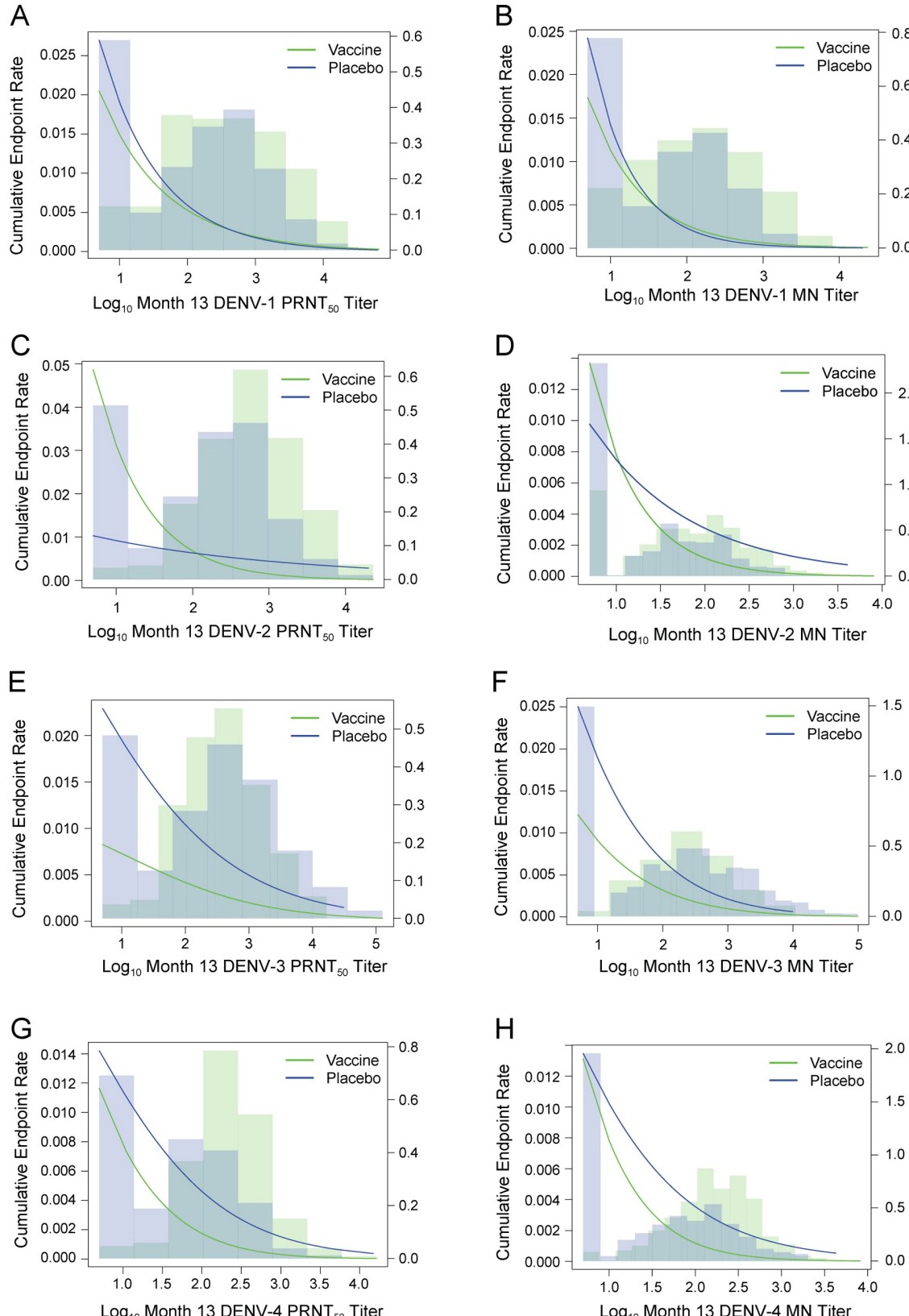

**Fig 6. Logistic regression estimates of cumulative endpoint rates for serotype-specific VCD and sampling weighted distributions of serotype-specific $\log_{10}$ nAb titers, in CYD14 and CYD15 together.** (A, C, E, G): Month 13 serotype-specific $PRNT_{50}$ titer, (B, D, F, H): Month 13 serotype-specific MN titer.

we found that estimated VE against DENV-Any was approximately 35% for vaccine recipients with no seroresponse (measured by the MN assay) at Month 13 and that it likewise increased similarly with average Month 13 MN titer in baseline-seronegative vs. baseline-seropositive subgroups in CYD14 and CYD15 9–16-year-old vaccine recipients (baseline serostatus determined by the $PRNT_{50}$ assay) (Fig 7C and 7D). Among CYD14 and CYD15 9–16-year-old vaccine recipients, VE estimates against DENV-Any at the median Month 13 average $PRNT_{50}$ titer of 392 were 77% for the baseline-seropositive subgroup and 68% for the baseline-seronegative subgroup; at a Month 13 average MN titer of the same value (392), VE estimates were 82% for the baseline-seropositive subgroup and 70% for the baseline-seronegative subgroup (Fig 7). Thus, stratification by baseline serostatus of VE estimates by Month 13 titer in CYD14 and CYD15 9–16-year-old vaccine recipients yields relatively similar results for baseline-seropositive vs. baseline-seronegative subgroups, regardless of which assay is used to measure Month 13 titer.

## Super-Learner prediction of individual DENV-Any outcomes

Using Super-Learner, an approach that selects the best weighted combination of prediction algorithms from multiple candidates [36], we next assessed whether/how MN and $PRNT_{50}$ titers helped predict individual-level VCD risk. Each included algorithm classified 9–16-year-old vaccine and placebo recipients in CYD14 and CYD15 as to whether they experienced DENV-Any VCD between Months 13 and 25. We identified best models for 4 covariate groups: 1) baseline demographic information, 2) demographic information+Month 13 MN titer, 3) demographic information+Month 13 $PRNT_{50}$ titer, and 4) demographic information +Month 13 MN titer+Month 13 $PRNT_{50}$ titer. S4 Table provides a complete list of the input variables (e.g. demographic variables, MN titer variables, $PRNT_{50}$ titer variables) used in each covariate group for the various supervised learning analyses, in addition to further information on the statistical learning algorithms in the Super-Learner library of estimators of the conditional probability of DENV-Any.

Classification accuracy using the best model identified by Super-Learner was overall better for the vaccine group, with CV-AUCs ranging from 0.61–0.84 (vaccine) vs. 0.54–0.74 (placebo) (Table 3). In both treatment groups the addition of Month 13 nAb titer (either assay) improved classification accuracy over demographic characteristics only, with CV-AUC increases ranging from 0.19–0.21. In the placebo group, the addition of Month 13 $PRNT_{50}$ titer data did not improve classification accuracy over that achieved with demographic +Month 13 MN titer data, nor did the addition of Month 13 MN titer data improve classification accuracy over that achieved with demographic+Month 13 $PRNT_{50}$ titer data (Table 3). In contrast, in the vaccine group slight improvement in classification accuracy was achieved by including both nAb titers with demographic information vs. including only one nAb titer and demographic information, particularly by additionally including Month 13 MN data when demographic+Month 13 $PRNT_{50}$ data were first considered. Specifically, in the vaccine group the CV-AUC for demographic+Month 13 $PRNT_{50}$ data alone was 0.79 (0.76–0.82), whereas it was 0.84 (0.82–0.87) for demographic+MN+$PRNT_{50}$ data (Table 3). Panels A and B of S2 Fig compare the CV-AUCs for all algorithms, including Super-Learner along with individual statistical algorithms such as standard logistic regression, applied using input variables demographics+MN+$PRNT_{50}$ data. The results show slight gains in classification accuracy for Super-

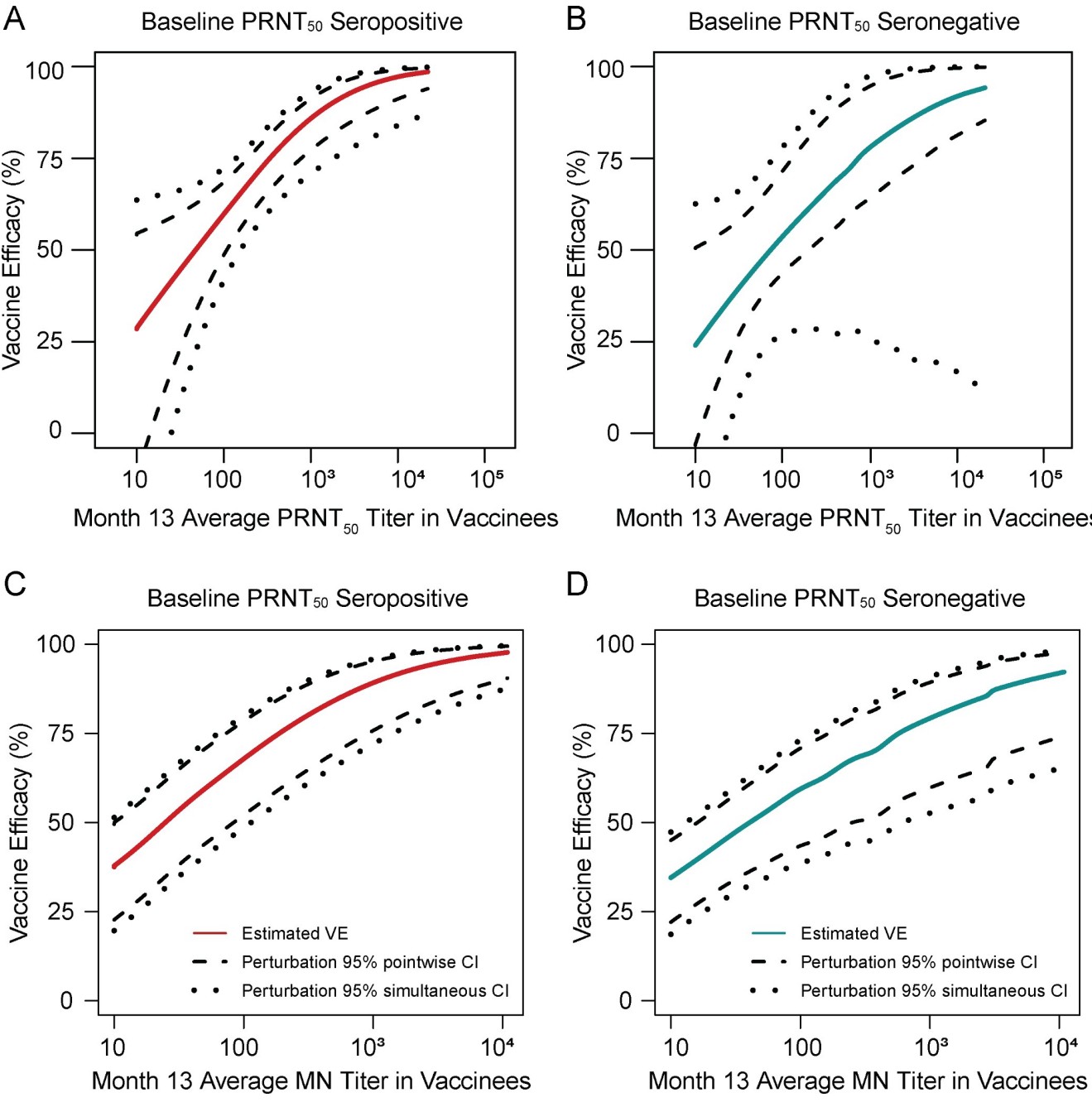

**Fig 7.** Estimated vaccine efficacy against DENV-Any by average $\log_{10}$ PRNT$_{50}$ (Panels A and B) or MN (Panels C and D) titer at Month 13 in baseline seropositive (Panels A and C) and baseline seronegative (Panels B and D) subgroups of CYD14 and CYD15 9–16-year-old vaccine recipients. 95% pointwise and simultaneous confidence intervals are also shown. Plots were generated using the Zhuang et al. method [35]. Baseline seropositive individuals were defined as being seropositive (PRNT$_{50}$ > 10) to at least one serotype and baseline seronegative individuals were defined as being seronegative (PRNT$_{50}$ ≤10) to all four serotypes. Panels A and B are reproduced with modification from Fig S15 in Moodie et al. [11] and are shown for comparison.

Learner compared to most of the individual algorithms, and large gains in classification accuracy for Super-Learner compared to the polymars, mean, and nnet algorithms. These results held true for both the placebo and vaccine groups. (Part B of S4 Table provides more information on the learning algorithms in the Super-Learner library of estimators of the conditional probability of DENV-Any).

**Table 3. DENV-Any classification accuracy for the best vaccine and placebo models for each input variable data set.**

| Variable Set Name | Treatment | Best Model | CV-AUC (95% CI)[a] |
|---|---|---|---|
| 1. Demo | Vaccine | SL.gam[b] | 0.61 (0.56–0.65) |
| | Placebo | SL.gam | 0.54 (0.50–0.59) |
| 2. Demo + MN | Vaccine | SuperLearner | 0.82 (0.79–0.85) |
| | Placebo | SuperLearner | 0.73 (0.69–0.76) |
| 3. Demo + PRNT$_{50}$ | Vaccine | SuperLearner | 0.79 (0.76–0.82) |
| | Placebo | SL.gam | 0.74 (0.70–0.77) |
| 4. Demo + MN + PRNT$_{50}$ | Vaccine | SuperLearner | 0.84 (0.82–0.87) |
| | Placebo | SL.gam | 0.73 (0.70–0.77) |

[a]CV-AUC is cross-validated area under the receiver operating characteristic curve, with 95% CI for the CV-AUC estimated by the method of Hubbard, Kherad-Pajouh, and van der Laan [37].

[b]SL.gam is a generalized additive model with smoothing splines for the neutralization titer variables.

Fig 8 shows the CV-ROC curves for the best-performing vaccine group models fit on each of the four input variable sets (Table 3). Panel B displays a magnified version of the CV-ROC curves with cross-validated false positive rate under 0.014 (the overall rate in placebo recipients) for the best vaccine group models. For very low false positive rates, the true positive rates are higher when demographic+MN+PRNT$_{50}$ information is included in the classification, intermediate performance is achieved by including nAb titer information from one assay (with potentially slightly better classification achieved by the addition of MN vs PRNT$_{50}$ titer), and worst performance is achieved when only demographic data are used.

S2 Fig, panel C shows the cross-validated estimated probabilities of DENV-Any by case-control status for the best-performing models for each covariate group, for the vaccine group. Cases are assigned higher predicted values of DENV-Any than controls when data from either nAb assay are included; this difference is greatest when MN+PRNT$_{50}$ information are both included, with potentially better prediction achieved by the addition of MN vs PRNT$_{50}$ titer. S5 Table, part A shows further information on one of the best-fitting and most easily interpretable models for the vaccine group based on demographic+MN+PRNT$_{50}$ information. The logistic regression model with 5 variables shows that greater age, Month 13 DENV-1 PRNT$_{50}$ titer, and Month 13 DENV-2 PRNT$_{50}$ titer are all inversely associated with risk of DENV-Any, whereas countries with higher rates of DENV-2 and DENV-3 have higher risk. S5 Table, part B shows analogous information for prediction based on demographic+MN information.

## Discussion

We conclude that the MN assay has equal or potentially even better utility than the PRNT$_{50}$ assay for defining CoRs, CoVEs, and individual-level predictors of DENV risk, particularly for individuals aged $\geq 9$ years. In both trials, average Month 13 MN titer was significantly inversely correlated with risk of VCD of any serotype and was even a slightly stronger correlate than average Month 13 PRNT$_{50}$ titer. As in our previous analysis of Month 13 PRNT$_{50}$ titers [11], high Month 13 MN titers were associated with high VE regardless of the subgroup (serostatus, age, study) and estimated VE increased with average Month 13 MN titer; moreover, like Month 13 PRNT$_{50}$ titer, no absolute threshold Month 13 MN titer was observed that was associated with 100% VE, classifying MN titers as a "relative correlate" [12]. We also found that for vaccine recipients with the lowest MN titers, VE may be closer to zero compared to vaccine recipients with the lowest PRNT$_{50}$ titers. This difference was most noticeable for VE against DENV-Any in 9–16-year-olds pooled across the two trials and for VE against DENV-4 in

A

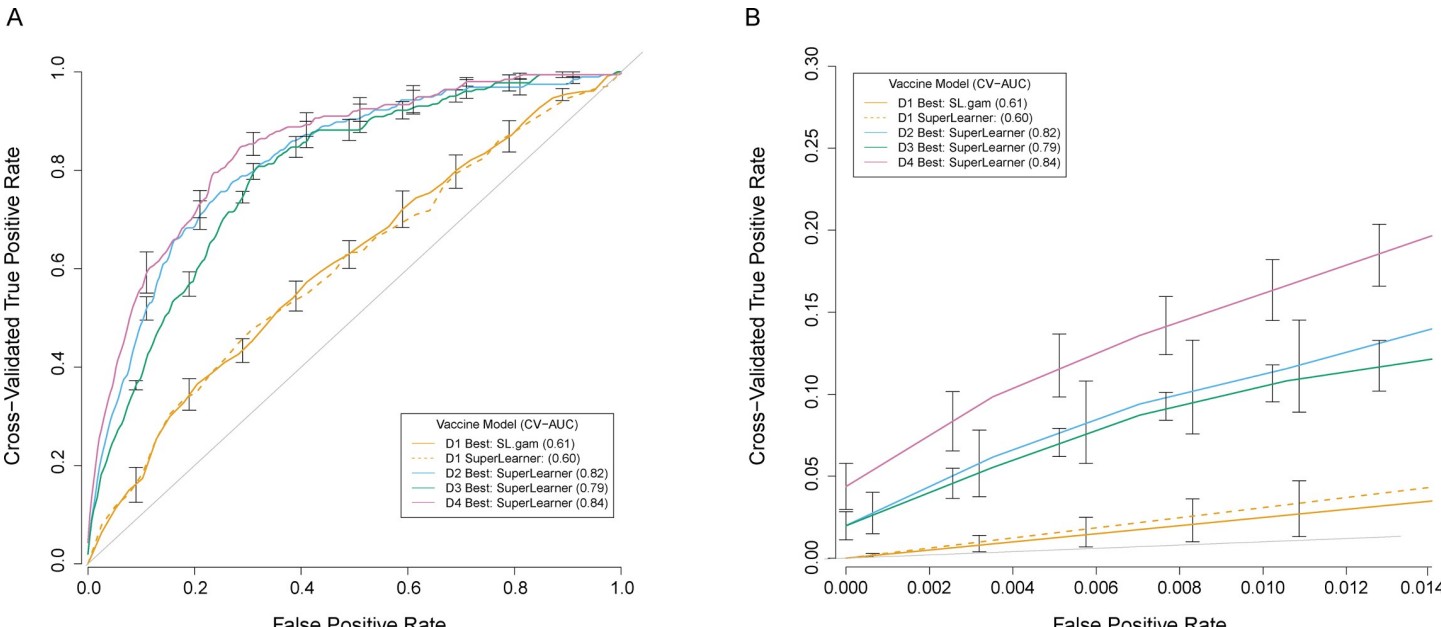

B

**Fig 8. Cross-validated ROC curves for the best vaccine group models fit on each of the four input variable sets.** D1 = dataset 1 (demographics alone); D2 = dataset 2 (demographic + Month 13 MN titer data); D3 = dataset 3 (demographic + Month 13 PRNT$_{50}$ titer data); D4 = dataset 4 (demographic + Month 13 MN + Month 13 PRNT$_{50}$ titer data). For comparison, the SuperLearner results for D1 are also plotted. Panel B shows a magnification of the lower left region of Panel A.

CYD15, suggesting that low MN titers may mark absent/low VE better than low PRNT$_{50}$ titers and that nAbs measured by the MN assay may mediate more of the VE in this age group. While there was insufficient precision to conclude greater mediation by nAbs measured by the MN assay, the data support that the MN assay is not worse. Finally, using Super-Learner, the best prediction of individual-level risk of VCD was achieved when both Month 13 PRNT$_{50}$ and MN titer data were included in the models, with potentially better performance achieved by the addition of MN vs PRNT$_{50}$ titer data. Considering these findings and the operational advantages of the MN assay, MN may be a suitable alternative assay to PRNT$_{50}$ in analyses of large-scale vaccine trials. A limitation of our analysis was that it was restricted to VCD risk and protection against VCD from Month 13 to Month 25. Future analyses would be needed to examine the utility of MN and PRNT titers in correlates analyses of other DENV endpoints and over longer follow-up periods.

While in CYD15 all titer readouts (serotype-specific and average) were significant CoRs for their corresponding VCD endpoints (matched-serotype VCD and DENV-Any VCD, respectively), it is unclear why some of the serotype-specific nAb titer readouts were stronger CoRs for their respective matched-serotype VCD endpoints than other serotype-specific nAb titer readouts were for their respective matched-serotype VCD endpoints. For example, for both assays, DENV-1 and DENV-3 titers tended to be less strong CoRs (albeit still significant CoRs) for serotype-matched VCD than DENV-2 and DENV-4 titers [PRNT$_{50}$ –DENV-1: 0.31 (0.23, 0.42); DENV-3: 0.41 (0.27, 0.64) vs DENV-2: 0.18 (0.12, 0.27); 0.22 (0.09, 0.51). MN–DENV-1: 0.23 (0.15, 0.33); DENV-3: 0.23 (0.13, 0.41) vs DENV-2: 0.14 (0.08, 0.26); DENV-4: 0.08 (0.03, 0.19)] (Table 1). Differences between in vitro systems for assessing antibody-mediated DENV neutralization, which use cultured cell lines and laboratory DENV strains, versus neutralization of circulating DENV viral variants in the human body, may be relevant. For instance, the PRNT$_{50}$ and MN assays assess neutralization of only one DENV strain per serotype. If participants are exposed to circulating viral variants that are neutralized less well (or better) than the

assayed strain, the obtained titer for that serotype will be less representative of how well nAbs in that participant's serum neutralize exposing viral variants. While it is generally assumed that nAb binding epitopes are conserved within serotypes, there is evidence supporting significant variation in neutralization across genotypes of a given serotype, particularly for DENV-1 and DENV-3 [38–40]. We speculate that a scenario in which contemporaneously circulating DENV-1 and DENV-3 strains are neutralized less well (or better) than the DENV-1 PUO-359 (isolated in 1980 in Thailand) and DENV-3 PaH881/88 (isolated in 1988 in Thailand) strains used in the PRNT$_{50}$ and MN assays (i.e. the parental DENVs of the respective recombinant vaccine viruses) could explain why DENV-1 and DENV-3 titers tended to be weaker (yet still significant) CoRs versus DENV-2 and DENV-4 titers.

The proportion of mismatched amino acid residues between the vaccine DENV inserts and the DENV sequences isolated from placebo group cases provides an assessment of the degree of match between circulating viral variants at the time of the trial and the vaccine strains, and may also be relevant to explain potential differences across serotypes in the strength of serotype-specific CoRs of matched-serotype VCD. We have previously analyzed these proportions by serotype and shown that, in CYD15, DENV-1 circulating strains were farthest from the DENV-1 vaccine insert, followed by DENV-2 circulating strains to the DENV-2 insert, DENV-3 circulating strains to the DENV-3 insert, and then DENV-4 circulating strains found to be closest to the DENV-4 vaccine insert [41]. The latter finding is consistent with DENV-4 titer being the strongest serotype-specific CoR of matched-serotype VCD in CYD15 (Table 1); however, these findings do not fully explain why DENV-2 titer tended to be a stronger CoR of DENV-2 VCD than DENV-1 titer was of DENV-1 VCD in CYD15.

Our analysis is distinguished from previous MN-based assay comparisons to the PRNT assay [15, 16, 42] by its much larger sample sizes. We found that MN and PRNT$_{50}$ titers had excellent correlation for all serotypes, with average titer Spearman correlation coefficients from pairwise plots equalling or exceeding 0.96, across both trials and time-points. Correlations were somewhat lower for DENV-4, perhaps due to the larger size of DENV-4 plaques, which may introduce more subjectivity into the counting, especially when plaques cluster together. Nonetheless, the DENV-4 correlations were higher than the $R^2$ value (0.672) reported in [42] and comparable to the Pearson correlation coefficient (0.84) reported in [15].

The two assays showed good agreement with respect to baseline serostatus classification, although the proportion of MN-/PRNT$_{50}$+ participants always slightly exceeded the proportion of MN+/PRNT$_{50}$- participants, across age groups and trials. Technical differences between the two assays must account for this observed difference in baseline seronegativity determination. Of note, the MN assay uses a higher virus input and lower serum volume in the neutralization reaction, thus having a much higher molar ratio of neutralizing epitopes to bind compared to the PRNT assay. Moreover, the MN assay does not use an overlay and thus the antibodies must continuously neutralize the virus, while the PRNT assay measures a one-hit neutralization event. The ability of antibodies to neutralize the virus is also affected by the replication rate of the individual viruses in the MN assay, while it is controlled in the PRNT assay.

Both PRNT$_{50}$ and MN are complex assays performed in specialized laboratories, typically for research/investigational purposes rather than patient care decisions. Thus, neither assay would be used for determining eligibility for CYD-TDV vaccination (e.g. in "pre-vaccination screening") [10]. However, the MN assay could be considered as an alternative test to determine eligibility of participation in future clinical trials of the CYD-TDV vaccine, with vaccination restricted to those testing positive. In this context, the MN assay may be theoretically somewhat advantageous to the PRNT$_{50}$ assay for excluding true dengue-seronegative individuals from vaccination, given that our findings suggest possible higher specificity of the MN assay to determine dengue seropositivity. Moreover, due to the possible lower sensitivity of the

MN assay, other vaccine candidates that base assessments of serostatus on MN may misclassify more true seropositives as seronegatives, potentially resulting in some degree of bias in sero-negative estimates with this or similar assays.

We next consider why individual-level classification accuracy, using the best model identified by Super-Learner for each of four different covariate groups, was relatively limited. Most participants in vaccine efficacy trials will not experience the VCD endpoint over the study follow-up timeframe, irrespective of vaccination, based solely on the epidemiology of exposure, infection, and symptomatic disease frequency. The relative rareness of the VCD endpoint presents a significant challenge in improving classification of individuals who will vs. will not experience the VCD endpoint. CYD-TDV vaccination may also impact risk differently depending on genetic/antigenic features of different variants within the same serotype or genotype [41], which could also explain the limited classification accuracy.

Overall, we conclude that Month 13 MN titer performs comparably to Month 13 $PRNT_{50}$ titer as a CoR and as a CoVE, supporting that the MN assay could be an alternative to the PRNT assay for assessing neutralizing antibody titers in immunogenicity studies, immune correlates studies, and immuno-bridging applications (e.g. validating a new vaccine lot).

## Supporting information

**S1 Table. Correlations between MN and $PRNT_{50}$ titers.**
(DOCX)

**S2 Table. Concordance with respect to baseline serostatus classification in the CYD14 and CYD15 trials for the MN and $PRNT_{50}$ assays.**
(DOCX)

**S3 Table. Logistic regression estimated odds ratios (ORs) (95% CIs) of matched-serotype VCD per $log_{10}$ increase in Month 13 neutralizing antibody titer as measured by each assay, pooling data from the CYD14 and CYD15 studies together.** For the $PRNT_{50}$ assay, the CoR analyses were performed on all CYD14 and CYD15 data, as reported in Moodie et al. [11], and for the MN assay, the CoR analyses were restricted to 9–16-year-olds from CYD14 and 15.
(DOCX)

**S4 Table. Distinct input variable sets used for learning algorithms and learning algorithms in the Super-Learner library of estimators of the conditional probability of DENV-Any.**
(DOCX)

**S5 Table. Model terms for the best interpretable model for the Vaccine group, for different data sets.**
(DOCX)

**S1 Fig.** Comparison of classification of CYD14 and CYD15 9–16-year-old immunogenicity subset (A, B) cases and controls or (B, C, E, F) controls as (A, B, C) dengue seronegative vs. (D, E, F) dengue seropositive at (A, D) baseline and at (B, C, E, F) Month 13 according to the PRNT50 (blue) or MN (red) assay. Seropositivity was defined as a titer $\geq 10$ for each individual serotype and as a titer $\geq 10$ of at least one serotype for the Average readout. Seronegativity was defined as a titer $< 10$ for each individual serotype and as a titer $< 10$ for all individual serotypes for the Average readout.
(TIF)

**S2 Fig.** Classification accuracy (A,B) of different algorithms using demographic + MN + PRNT50 data and cross-validated estimated probabilities of DENV-Any by case-control

status (C). (A, B): CV-AUC values for classification accuracy of different algorithms using demographic + MN + PRNT50 data as to whether each participant experienced DENV-Any VCD between Months 13 and 25 are shown for (A) the vaccine group and (B) the placebo group for the combined CYD14 and CYD15 9-16-year-old cohort. (C) Cross-validated estimated probabilities of DENV-Any in the vaccine group by case-control status for the best-performing models for each covariate group for the combined CYD14 and CYD15 9-16-year-old cohort.
(TIF)

**S1 Text. Case-cohort sampling design for measurement of Month 13 MN titers in CYD14 and CYD15 participants.**
(DOCX)

## Acknowledgments

We thank the participants, their parents/guardians, and the investigators of the CYD14 and CYD15 trials.

## Author Contributions

**Conceptualization:** Matthew Bonaparte, Saranya Sridhar, Carlos A. DiazGranados, Peter B. Gilbert.

**Data curation:** Matthew Bonaparte, Lingyi Zheng.

**Formal analysis:** Youyi Fong, Matthew Bonaparte, Zoe Moodie, Michal Juraska, Ying Huang, Brenda Price, Yingying Zhuang, Jason Shao, Lingyi Zheng, Laurent Chambonneau.

**Funding acquisition:** Carlos A. DiazGranados, Peter B. Gilbert.

**Investigation:** Matthew Bonaparte, Lingyi Zheng.

**Methodology:** Youyi Fong, Matthew Bonaparte, Michal Juraska, Ying Huang, Brenda Price, Saranya Sridhar, Carlos A. DiazGranados, Peter B. Gilbert.

**Project administration:** Zoe Moodie, Saranya Sridhar, Carlos A. DiazGranados, Peter B. Gilbert.

**Software:** Youyi Fong, Michal Juraska.

**Supervision:** Matthew Bonaparte, Robert Small, Saranya Sridhar, Carlos A. DiazGranados, Peter B. Gilbert.

**Visualization:** Lindsay N. Carpp, Youyi Fong, Matthew Bonaparte, Zoe Moodie, Michal Juraska, Ying Huang, Brenda Price, Yingying Zhuang, Jason Shao.

**Writing – original draft:** Lindsay N. Carpp, Youyi Fong, Peter B. Gilbert.

**Writing – review & editing:** Lindsay N. Carpp, Youyi Fong, Matthew Bonaparte, Zoe Moodie, Michal Juraska, Ying Huang, Brenda Price, Yingying Zhuang, Jason Shao, Lingyi Zheng, Laurent Chambonneau, Robert Small, Saranya Sridhar, Carlos A. DiazGranados, Peter B. Gilbert.

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
