## [Decision Letter · Decision Letter 0]

9 Mar 2020

PONE-D-19-30338

Microneutralization Assay Titer Correlates Analysis in Two Phase 3 Trials of the CYD-TDV Tetravalent Dengue Vaccine in Asia and Latin America

PLOS ONE

Dear Prof Gilbert,

Thank you for submitting your manuscript to PLOS ONE. After careful consideration, we feel that it has merit but does not fully meet PLOS ONE’s publication criteria as it currently stands. Therefore, we invite you to submit a revised version of the manuscript that addresses the reviewers' points below raised during the review process.

We would appreciate receiving your revised manuscript by Apr 23 2020 11:59PM. To enhance the reproducibility of your results, we recommend that if applicable you deposit your laboratory protocols in protocols.io, where a protocol can be assigned its own identifier (DOI) such that it can be cited independently in the future. For instructions see: http://journals.plos.org/plosone/s/submission-guidelines#loc-laboratory-protocols

We look forward to receiving your revised manuscript.

Kind regards,

Ray Borrow, Ph.D., FRCPath

Academic Editor

PLOS ONE

Journal Requirements:

"I have read the journal's policy and the authors of this manuscript have the following competing interests: MB, LZ, LC, RS, SS, and CDG are employees of Sanofi Pasteur. LNC, YF, ZM, MJ, YH, BP, YZ, JS, and PBG received a contract from Sanofi Pasteur to conduct the statistical analysis work and submit the results for publication. Sanofi Pasteur is the manufacturer of the CYD-TDV vaccine (Dengvaxia).".

 i) Please confirm that this does not alter your adherence to all PLOS ONE policies on sharing data and materials, by including the following statement: "This does not alter our adherence to  PLOS ONE policies on sharing data and materials.” (as detailed online in our guide for authors http://journals.plos.org/plosone/s/competing-interests). If there are restrictions on sharing of data and/or materials, please state these. Please note that we cannot proceed with consideration of your article until this information has been declared.

 ii) Please include your updated Competing Interests statement in your cover letter; we will change the online submission form on your behalf.

Reviewers' comments:

Reviewer's Responses to Questions

**Comments to the Author**

1. Is the manuscript technically sound, and do the data support the conclusions?

Reviewer #1: Yes

Reviewer #2: Yes

Reviewer #3: Yes

Reviewer #4: Partly

2. Has the statistical analysis been performed appropriately and rigorously? 

Reviewer #1: Yes

Reviewer #2: Yes

Reviewer #3: Yes

Reviewer #4: Yes

3. Have the authors made all data underlying the findings in their manuscript fully available?

Reviewer #1: Yes

Reviewer #2: Yes

Reviewer #3: No

Reviewer #4: Yes

4. Is the manuscript presented in an intelligible fashion and written in standard English?

Reviewer #1: Yes

Reviewer #2: Yes

Reviewer #3: Yes

Reviewer #4: Yes

5. Review Comments to the Author

Reviewer #1: In this manuscript, the authors have developed a high throughput micro-neutralization assay to assess neut. titers of dengue viruses post immunization with Sanofi Pasteur’s tetravalent dengue vaccine. While the PRNT assay is considered the gold standard assay, it is laborious and there are challenges associated with performing the assay in laboratories. Using samples from month 13 and at baseline from the CYD 14 and 15 clinical trials, the authors compare titers using the MN and PRNT50 assays. They find strong correlations between high titers at M13 and vaccine efficacy regardless of serotype. Give the high throughput nature of the MN assay, it certainly would be valuable for vaccine manufacturers to assess a main correlate of protection.

The MN is similar to the PRNT assays and does not solve challenges such as the use of lab strains of virus, cultured cell lines for testing. Other MN assays have been published before and the advantages of this assay are the relatively high throughput nature of the assay and the number of serum samples tested which lend further validity to the assay. The authors have carefully layed out the differences in PRNT50 and MN assays including higher virus input, lower serum amount and antibodies being continuously able to neutralize virus in the reaction.

Overall, the data provided, the large number of samples tested in the MN assay and the stringent comparisons made to the traditional PRNT50 assays lend confidence that the assay should be used to test additional end points.

Reviewer #2: The manuscript by Carpp et al reported the development of new microneutralization (MN) assay and compared the MN titers with a validated 50% plaque reduction neutralization test (PRNT50) in analysis of the correlates in tow Phase 3 clinical trials of the CYDTDV Tetravalent Dengue Vaccine in Asia and Latin America. The results demonstrated that the neutralizing titers assessed by MN and PRNT50 assays had good correlates. The high-throughput MN assay can be useful to for assessing neutralizing antibody responses induced by Dengue vaccines and infections to evaluate the correlates of risk and vaccine efficacy. The manuscript and data are well-organized.

Reviewer #3: The manuscript develops and validates the MN assay in light of the PRNT assay, with respect to dengue vaccine development. The authors consider a case-cohort design. The paper is nicely written, provides significant details, and the statistical analyses is rigorous and appropriate. I have some additional commens/clarifications:

1. In Page 6 (Middle), the authors state "....first 2 to 4 months of each trial". Is this trial registered with ClinicalTrials.gov and has a NCT number? If not, why is it then called a clinical trial? Any prior trial? Details needed for a smoother reading experience.

2. Any justification behind the use of the SuperLearner package? Has it been established that this ensemble machine-learning technique almost always produces better prediction and discriminatory performances in this specific field of research (dengue vaccime development)? Maybe a comparison with a standard statistical model, and further illustration of the actual gain would be worthwhile. I see a comparison with SL.gam; can the authors specify what is that? I assume it is some generalized additive model, and certainly not a standard regression.

3. It was hard to find the complete list of covariates fed into the SuperLearner. It would be better to provide the list somewhere during model fitting.

4. Inverse probability of weighting (line 154, page 8) require a citation.

5. Line 155, page 8: "Models were built...". What models? Write clearly.

6. Line 156, page 8: expand CV-AUC, with a reference.

Reviewer #4: Carpp and Fong et al. compare how well a high-throughput microneutralization assay compares to the gold-standard PRNT50 assay as a correlate of risk and correlate of vaccine efficacy against virologically confirmed dengue in the CYD-TDV Phase 3 vaccine trials. Overall, this manuscript provides valuable information on comparison of the two assays in the context of a vaccine trial, as well as additional novel scientific investigations into dengue vaccine efficacy.

Major comments:

Methods:

- There is very little information provided about the MN. Key useful details that should be included for the MN include how much virus is added to wells, how long virus is allowed to replicate in cells before the assay is terminated, how the assay is terminated, etc. The authors mention some differences between the MN and the PRNT in the discussion. However, actual information on the two assays is not detailed in the manuscript.

- The methods section on the immune correlates analyses only includes a reference to a previous article. There is no limit on words for this manuscript. It would be helpful to the reader to provide a brief description of the immune correlates analyses in the methods section.

Results:

- It is unusual that there are entire paragraphs on results that are only shown in supplemental tables and figures. Why not just include these data as additional manuscript figures and tables? Especially Table S3 and Fig. S2.

- Line 351-352: “the DENV-2 and DENV-4 correlates of risk were significantly modified by treatment group, indicating departure from the third criterion”. While this is true of DENV2 and DENV4, the effects go in opposite directions and are different in important ways. The treatment contributes to a higher cumulative endpoint rate for DENV2 at low titers but a lower cumulative endpoint rate for DENV4 across titers. Also, there is a significant elevated risk of the treatment group in Model 4 for DENV2 in Table S4. These important findings should be mentioned in the text.

- Line 352-354: "Together, these results show that Month 13 PRNT50 titer and Month 13 MN titer are consistent with the Prentice criteria for DENV-1 but not for the other serotypes." This is quite interesting. Why is this as supplemental figure (Fig. S2)?

- Line 360-361: "We previously showed that estimated VE against DENV-Any was approximately 25% for vaccine recipients with no seroresponse at Month 13…” Presumably, no seroresponse means titers <10? The model estimates for the undetectable titers (<10) are not shown in the figure, nor in the original figure S15 of Moodie et al. 2017. However, the <10 value is shown on the x-axis. The figures appear to only show titers from a value of 10, which is detectable. Is this just a plotting issue? On the microneut panel in Fig. 6, the x-axis only goes to 10, even though there are individuals with MN values <10.

- Line 368-370: "At an average MN (PRNT50) Month 13 titer of 1000, VE estimates were 88% (85%) for baseline-seropositive vaccine recipients and 78% (76%) for baseline-seronegative vaccine recipients (Figure 6)." Why state the VE at this high a titer value? Based on the titer distributions, it was rare for individuals to have that high of titers in the trial even among controls. Perhaps it would be better to report VE based on the median titer observed.

- Part B Table S6: the model shows some very strong significant OR >>>1. (e.g. 52, 79, 830). What does this mean? Some of the effects seem to be for interaction terms? It is very difficult to interpret what this means without a description of what the terms are.

Discussion:

- Paragraph, 470: “It is unclear why nAb titer readouts did not perform equally well across serotypes as CoRs for their matched-serotype VCD endpoints.” Is this paragraph referring to Table 1? I thought the non-significant effects were DENV3 and DENV4? This paragraph mentions DENV1 and DENV3?

- Line 517: "We next consider why individual-level classification accuracy using the different variable input." Is this referring to the super-learner algorithm?

Minor comments:

- Line 60: "Estimated VE against these two endpoints was negative in baseline-seronegative individuals”. By negative, you mean had a negative vaccine efficacy, meaning worse off? This is unclear as written.

- Line 156: Please write out CV-AUC (cross validated area under the curve) the first time it appears.

- "Lack of Month 13 sero-response at Month 13 approached zero for both assays (≤ 2%) for DENV-3 and DENV-4 (P>0.05 for both), but significant discordance remained between the MN and PRNT50 results for DENV-1 and DENV-2 (P<0.01 for both) (S1 Fig, panel C)." This sentence is confusing as written.

- Line 232-234: "As one log10 increase in PRNT50 titer approximately equaled one log10 increase in MN titer (Figure 1), we used OR for comparing the two nAb readouts as CoRs.” Why not directly quantify this relationship? Why say approximately one log10 increase here?

- Fig. S2: the cumulative endpoint rate for Fig S2 C and D are quite different (DENV2): 0.05 for the PRNT vs. 0.014 for MN assay? Other y-axes of the PRNT50 vs. MN panels in this figure are more consistent.

- Fig. 7: no color is shown in the legend for for D2 model panel A. I assume it should be blue?

6. PLOS authors have the option to publish the peer review history of their article (what does this mean?). If published, this will include your full peer review and any attached files.

Reviewer #1: Yes: Anuja Mathew

Reviewer #2: No

Reviewer #3: No

Reviewer #4: No

---

## [Author Response · Author response to Decision Letter 0]

12 Apr 2020

Journal Requirements:

Response: We have ensured that all PLOS ONE style requirements are met in the revision.

"I have read the journal's policy and the authors of this manuscript have the following competing interests: MB, LZ, LC, RS, SS, and CDG are employees of Sanofi Pasteur. LNC, YF, ZM, MJ, YH, BP, YZ, JS, and PBG received a contract from Sanofi Pasteur to conduct the statistical analysis work and submit the results for publication. Sanofi Pasteur is the manufacturer of the CYD-TDV vaccine (Dengvaxia).".

i) Please confirm that this does not alter your adherence to all PLOS ONE policies on sharing data and materials, by including the following statement: "This does not alter our adherence to PLOS ONE policies on sharing data and materials.” (as detailed online in our guide for authors http://journals.plos.org/plosone/s/competing-interests). If there are restrictions on sharing of data and/or materials, please state these. Please note that we cannot proceed with consideration of your article until this information has been declared.

 ii) Please include your updated Competing Interests statement in your cover letter; we will change the online submission form on your behalf.

Response: We have included the statement above in our revised Competing Interests section.

Reviewer #1: In this manuscript, the authors have developed a high throughput micro-neutralization assay to assess neut. titers of dengue viruses post immunization with Sanofi Pasteur’s tetravalent dengue vaccine. While the PRNT assay is considered the gold standard assay, it is laborious and there are challenges associated with performing the assay in laboratories. Using samples from month 13 and at baseline from the CYD 14 and 15 clinical trials, the authors compare titers using the MN and PRNT50 assays. They find strong correlations between high titers at M13 and vaccine efficacy regardless of serotype. Give the high throughput nature of the MN assay, it certainly would be valuable for vaccine manufacturers to assess a main correlate of protection.

The MN is similar to the PRNT assays and does not solve challenges such as the use of lab strains of virus, cultured cell lines for testing. Other MN assays have been published before and the advantages of this assay are the relatively high throughput nature of the assay and the number of serum samples tested which lend further validity to the assay. The authors have carefully layed out the differences in PRNT50 and MN assays including higher virus input, lower serum amount and antibodies being continuously able to neutralize virus in the reaction.

Overall, the data provided, the large number of samples tested in the MN assay and the stringent comparisons made to the traditional PRNT50 assays lend confidence that the assay should be used to test additional end points.

Reviewer #2: The manuscript by Carpp et al reported the development of new microneutralization (MN) assay and compared the MN titers with a validated 50% plaque reduction neutralization test (PRNT50) in analysis of the correlates in tow Phase 3 clinical trials of the CYDTDV Tetravalent Dengue Vaccine in Asia and Latin America. The results demonstrated that the neutralizing titers assessed by MN and PRNT50 assays had good correlates. The high-throughput MN assay can be useful to for assessing neutralizing antibody responses induced by Dengue vaccines and infections to evaluate the correlates of risk and vaccine efficacy. The manuscript and data are well-organized.

Reviewer #3: The manuscript develops and validates the MN assay in light of the PRNT assay, with respect to dengue vaccine development. The authors consider a case-cohort design. The paper is nicely written, provides significant details, and the statistical analyses is rigorous and appropriate. I have some additional commens/clarifications:

Response: Thank you for the positive comments and feedback.

1. In Page 6 (Middle), the authors state "....first 2 to 4 months of each trial". Is this trial registered with ClinicalTrials.gov and has a NCT number? If not, why is it then called a clinical trial? Any prior trial? Details needed for a smoother reading experience.

Response: The CYD14 and CYD15 trials were both registered with ClinicalTrials.gov and each has an NCT number. This information is included in the “CYD14 and CYD15” subsection of “Materials and methods”. For additional clarity, we have added “ClinicalTrials.gov ID” in front of each NCT number (new text in this revision is underlined):

“In harmonized designs, healthy children and adolescents aged 2-14 (CYD14; ClinicalTrials.gov ID NCT01373281 [6]) or 9-16 (CYD15; ClinicalTrials.gov ID NCT01374516 [7]) were randomized (2:1) to vaccine or placebo, with randomization stratified by age group and site.” (lines 97-98)

2. Any justification behind the use of the SuperLearner package? Has it been established that this ensemble machine-learning technique almost always produces better prediction and discriminatory performances in this specific field of research (dengue vaccine development)? Maybe a comparison with a standard statistical model, and further illustration of the actual gain would be worthwhile. I see a comparison with SL.gam; can the authors specify what is that? I assume it is some generalized additive model, and certainly not a standard regression.

Response: 

Any justification behind the use of the SuperLearner package?

Super-Learner possesses an oracle property, in that it selects a learner as good asymptotically as any individual learner in the specified ensemble of learners. The fact its model selection is based on double-nested cross-validation and is implemented in press-button fully pre-specified fashion makes it objective. Moreover, its ability to easily incorporate a large library of learners allows it to have good performance in accuracy and precision of models, as found in many simulation studies and data applications (1, 2). Our previous experience using the Super-Learner R package includes assessment of varicella zoster virus-specific glycoprotein-based enzyme-linked immunosorbent assay (gpELISA) antibody titer as a predictor of herpes zoster in the Zostavax Efficacy and Safety Trial (3), assessment of dozens of antibody and T cell markers as predictors of HIV infection in the HVTN 505 HIV vaccine efficacy trial (4), and assessment of gp160 amino acid sequence features as predictors of whether HIV-1 Envelope pseudoviruses are resistant to neutralization by the monoclonal antibody VRC01 (2). In another application of Super-Learner to the CYD14 and CYD15 trials, we found that both logistic regression and Super-Learner were highly predictive of pre-vaccination dengue PRNT50 serostatus (5). 

Has it been established that this ensemble machine-learning technique almost always produces better prediction and discriminatory performances in this specific field of research (dengue vaccine development)?

For application to the dengue vaccine efficacy trials CYD14 and CYD15 in particular, Super-Learner is always either the best predictive model or close to the best predictive model, and because the classification accuracy is estimated by double-nested cross-validation, these comparisons are fair. In a statistical methods paper (6), we studied the Super-Learner package by applying it to the CYD14 and CYD15 trials as well as studying its operating characteristics in simulation studies, which verified that its general properties seemed to carry over to the dengue vaccine application setting. However, Super-Learner does not always provide best-predictive performance on a specific data set, as can be seen from the results in Table 3, where Super-Learner provided best performance for 4 of the 8 data analyses, whereas a generalized additive model (SL.gam) provided best performance for the other 4 of 8 data analyses. The Super-Learner model was always close to the best performing model as judged by cross-validated area under the ROC curve (CV-AUC). 

Maybe a comparison with a standard statistical model, and further illustration of the actual gain would be worthwhile.

We agree that such a comparison is worthwhile to the reader. S2 Fig provides such a comparison. It compares the classification accuracy (as to whether a participant experienced DENV-Any VCD between Months 13 and 25, using demographic + MN + PRNT50 data) of different algorithms, including Super-Learner and individual statistical algorithms such as logistic regression using all variables (SL.glm) and logistic regression with best model selected by step-wise model-selection (SL.step). 

We have added the following text (underlined) to the revision:

Panels A and B of S2 Fig compare the CV-AUCs for all algorithms, including Super-Learner along with individual statistical algorithms such as standard logistic regression, applied using input variables demographics+MN+PRNT50 data. The results show slight gains in classification accuracy for Super-Learner compared to most of the individual algorithms, and large gains in classification accuracy for Super-Learner compared to the polymars, mean, and nnet algorithms. These results held true for both the placebo and vaccine groups. (Part B of S4 Table provides more information on the learning algorithms in the Super-Learner library of estimators of the conditional probability of DENV-Any). (lines 502-509)

I see a comparison with SL.gam; can the authors specify what is that? I assume it is some generalized additive model, and certainly not a standard regression.

As stated in part B of S4 Table, “Learning Algorithms in the Super-Learner Library of Estimators of the Conditional Probability of DENV-Any”, all algorithm type names are built-in algorithms in the Super-Learner R package available at CRAN, and have documentation within the package. However, your comment makes us realize that the tables in the manuscript need more complete annotation of the meaning of the algorithm types. We have added the following information to the tables as appropriate:

To S4 Table: 

“b SL.mean is a base reference model using no input variables; SL.glm is a logistic regression model fit to all input variables; SL.glm.interaction is a logistic regression model including all input variables together with all pairwise-interaction variables; SL.step is a logistic regression model with step-wise model selection with best model selected by the AIC criterion; SL.glmnet is the lasso that includes variables with non-zero estimated coefficients in the default implementation of SL.glmnet that optimizes the tuning parameter via cross-validation; SL.bayesglm is Bayesian logistic regression; SL.gam is a generalized additive model with smoothing splines for the neutralization titer variables; SL.nnet is a neutral network; SL.polymars is multivariate adaptive polynomial spline regression.”

To Table 3: 

“SL.gam is a generalized additive model with smoothing splines for the neutralization titer variables.”

3. It was hard to find the complete list of covariates fed into the SuperLearner. It would be better to provide the list somewhere during model fitting.

Response: Thank you for raising this point. The complete list of the input variables (e.g. demographic variables, MN titer variables, PRNT50 titer variables) used in the various supervised learning analyses is given in Part A of S4 Table. We have expanded the description of S4 Table in the text as follows:

“S4 Table provides a complete list of the input variables (e.g. demographic variables, MN titer variables, PRNT50 titer variables) used in each covariate group for the various supervised learning analyses, in addition to further information on the statistical learning algorithms in the Super-Learner library of estimators of the conditional probability of DENV-Any.” (lines 483-487)

4. Inverse probability of weighting (line 154, page 8) require a citation.

Response: We have added one (“Inverse probability of censored weighting [31] was employed….”) (line 199). 

5. Line 155, page 8: "Models were built...". What models? Write clearly.

Response: We have revised the text as follows (added text is underlined): 

“Models of the conditional probability of DENV-Any occurrence by the Month 25 visit were built separately for the vaccine and placebo groups using four input variable sets…” (lines 200-203)

6. Line 156, page 8: expand CV-AUC, with a reference.

Response: We have expanded CV-AUC and added a reference.

“…aiming to maximize the cross-validated area under the receiver operating characteristic curve (CV-AUC) [32].” (lines 202-203)

Reviewer #4: Carpp and Fong et al. compare how well a high-throughput microneutralization assay compares to the gold-standard PRNT50 assay as a correlate of risk and correlate of vaccine efficacy against virologically confirmed dengue in the CYD-TDV Phase 3 vaccine trials. Overall, this manuscript provides valuable information on comparison of the two assays in the context of a vaccine trial, as well as additional novel scientific investigations into dengue vaccine efficacy.

Response: Thank you for the positive feedback.

Major comments:

Methods:

- There is very little information provided about the MN. Key useful details that should be included for the MN include how much virus is added to wells, how long virus is allowed to replicate in cells before the assay is terminated, how the assay is terminated, etc. The authors mention some differences between the MN and the PRNT in the discussion. However, actual information on the two assays is not detailed in the manuscript.

Response: Please see the text added in “MN assay” in Methods (lines 138-164):

“Briefly, 2-fold serial dilutions of serum samples (starting at 1:5 dilution) were incubated with an equivalent volume of a constant challenge dose of virus (200 TCID50 per well for each serotype) and incubated for 90 minutes at 37°C. A separate virus titration plate was prepared to determine the 50% tissue-culture infective dose (TCID50). After neutralization, the serum-virus mixture was added to pre-seeded Vero cell monolayers in 96-well plates and an additional 100 �l of cell culture medium was added without removal of the virus inoculum after adsorption. The plates were incubated at 37°C for either 4-5 days (depending on the virus serotype). The target virus challenge dose and days of incubation post-infection were determined for each serotype in order to provide an optimal signal-to-noise ratio during the ELISA steps. After the incubation period was complete, the cell culture medium was removed from the plates. The cells were then fixed with 80% acetone and incubated at room temperature for 10-15 min, followed by blocking with 5% non-fat dry milk in PBS-Tween-20 wash buffer. Dengue serotype-specific monoclonal antibodies were added, followed by anti-mouse IgG HRP congugate and TMB substrate. The reaction was stopped with 2N sulfuric acid and the optical density (OD) of each well at 450 nm (650 nm as the reference wavelength) was measured using a SpectraMax 384 microplate reader with SoftMax Pro software version 6.5.1.

The 50% neutralization titer of the test serum sample against each serotype was defined as the reciprocal of the test serum dilution for which the virus infectivity was reduced by 50% relative to the challenge virus dose (without any antibodies) introduced into the assay and was calculated using the formula: [(Average OD of Virus Control - Average OD of Cell Control)/2 + Average OD of Cell Control]. The MN titer for each test sample was interpolated by calculating the slope and intercept using the last dilution with an OD below the 50% neutralization point and the first dilution with an OD above the 50% neutralization point to determine the MN titer using the formula: [MN Titer = (50% neutralization point - intercept)/slope]. Neutralization titers are presented as continuous values. For both assays, the lower limit of quantitation was 10; values below this were set to 5. The average titer is the average of each participant’s four serotype-specific log10 titers.” 

- The methods section on the immune correlates analyses only includes a reference to a previous article. There is no limit on words for this manuscript. It would be helpful to the reader to provide a brief description of the immune correlates analyses in the methods section.

Response: We have added additional detail (underlined text) to the “Immune correlates” subsection in the “Materials and methods” section, as shown below:

 “Month 13 MN titers were assessed as CoRs and CoVEs as in [11]. In brief, the CoR analyses were performed in each of the vaccine and placebo groups separately, relating VCD risk to a given Month 13 MN titer variable with a logistic regression model that accounted for the case-cohort sampling design [22] and adjusted for age, sex, and country. Results are reported as odds ratios of DENV-Any, DENV-1, DENV-2, DENV-3, and DENV-4 VCD per log10 increase in Month 13 nAb titer. P values for testing DENV-1, DENV-2, DENV-3, and DENV-4 nAb titer as a CoR were adjusted across the 4 serotypes using family-wise error rate (Holm-Bonferroni [23]) and false-discovery rate (Q values [24]) adjustment, separately for each treatment group and each trial. All P values and Q values are 2-sided. The CoVE analyses were performed using the VE curve-effect modification framework [25-27]. This framework assesses how VE changes over subgroups of vaccine recipients, where subgroups are defined by Month 13 nAb titers. The analyses used the Juraska et al. method [28], employed with hinge point logit linear models [29] when there was sufficient data, if not, linear logistic regression models were used. Advantages of the hinge point models are summarized in reference [29]. VE curves were estimated with pointwise and simultaneous bootstrap-based Wald 95% confidence intervals (CIs).” (lines 173-188)

Results:

- It is unusual that there are entire paragraphs on results that are only shown in supplemental tables and figures. Why not just include these data as additional manuscript figures and tables? Especially Table S3 and Fig. S2.

Response: Thank you for the helpful suggestion. We have moved Table S3 to the main text (Table 2 in the revised manuscript) and Fig. S2 to the main text (Fig 6 in the revised manuscript). 

- Line 351-352: “the DENV-2 and DENV-4 correlates of risk were significantly modified by treatment group, indicating departure from the third criterion”. While this is true of DENV2 and DENV4, the effects go in opposite directions and are different in important ways. The treatment contributes to a higher cumulative endpoint rate for DENV2 at low titers but a lower cumulative endpoint rate for DENV4 across titers. Also, there is a significant elevated risk of the treatment group in Model 4 for DENV2 in Table S4. These important findings should be mentioned in the text.

Response: This is an interesting point. Care must be taken in making comments about the vaccine effect in subgroups defined by DENV-2 titer measured 13 months after randomization, as the models employed in Fig. S2 (Fig 6 in the revised manuscript) and Table S3 (Table 2 in the revised manuscript) do not measure a causal vaccine effect. When studying the vaccine effect in such post-vaccination subgroups there are concerns for post-randomization selection bias (7). This is why we instead focused on reporting correlate of VE curves (as reported in Fig 5), an approach designed to avoid potential post-randomization selection bias by estimating a causal vaccine effect across subgroups defined by Month 13 DENV-2 titer if assigned vaccine (which is a counterfactual variable for individuals assigned to the placebo group). In our previous work by Moodie et al. (8), which assessed correlates of risk and correlates of vaccine efficacy restricting to the PRNT50 assay, we studied whether VE against DENV-2 might be below zero at low Month 13 DENV-2 PRNT50 titers of vaccine recipients. We quote from our Moodie et al. work: “The DENV-2 VE curves across the complete age range in each trial (Supplementary Figures S8 and S9) suggest possible negative VE for vaccine recipients without anti-DENV-2 titers (i.e., PRNT50 below the LLOQ). However, the simultaneous 95% confidence bands for DENV-2 VE include 0%, and an inference of negative VE is based on a very small number of vaccine-recipient DENV-2 cases (0 of 87 in CYD14 and 5 of 264 in CYD15).” For convenience, the DENV-2 panels in Figures S8 and S9 from Moodie et al. are included below. 

Panel B (DENV-2 panel) from Figure S8 in Moodie et al. 2018 JID: Estimated vaccine efficacies in CYD14 for all age groups against serotype-specific dengue endpoints between Months 13 and 25 by homologous titers at Month 13 with 95% pointwise and simultaneous confidence intervals and histograms of the homologous titers in the vaccine group for the Month 13 at-risk cohort.

Panel B (DENV-2 panel) from Figure S9 in Moodie et al. 2018 JID: Estimated vaccine efficacies in CYD15 for all age groups against serotype-specific dengue endpoints between Months 13 and 25 by homologous titers at Month 13 with 95% pointwise and simultaneous confidence intervals and histograms of the homologous titers in the vaccine group for the Month 13 at-risk cohort.

Examining Fig 6, there is less evidence for potential negative VE based on DENV-2 MN titer than based on DENV-2 PRNT50 titer, such that the concern for vaccine-enhancement at low titers was already addressed in Moodie et al. We added the following text (lines 417-428):

“Regarding point (3), the cumulative endpoint rates of DENV-2 VCD by Month 13 DENV-2 PRNT50 titer (Panel C of Fig 6) suggest that CYD-TDV vaccination could have increased DENV-2 VCD risk at lowest Month 13 DENV-2 PRNT50 titers. Moodie et al. [11] previously addressed this issue, noting that simultaneous 95% confidence bands for DENV-2 VE include 0%, and an inference of vaccine-increased risk is based on a very small number of vaccine-recipient DENV-2 cases (0 DENV-2 cases among the 87 vaccine-recipients with no Month 13 PRNT50 DENV-2 seroresponse in CYD14 and 5 DENV-2 cases among the 264 vaccine-recipients with no Month 13 PRNT50 DENV-2 seroresponse in CYD15, with “no Month 13 PRNT50 DENV-2 seroresponse” defined as Month 13 PRNT50 DENV-2 titer < 10). Panel D of Fig 6 shows less concern for potential vaccine-increased DENV-2 VCD risk for individuals with lowest DENV-2 titers based on the MN assay, given that the vaccine and placebo curves are more similar in the left-tail of the plot.” 

Addressing the reviewer’s related comment about DENV-4, we have also added the following text (lines 429-432):

“For DENV-4, cumulative endpoint rates decreased with increasing Month 13 serotype-matched titers in both treatment groups, with low cumulative DENV-4 rates at low titers (Panels G and H of Fig 6), and a borderline significant result that the rate was lower in the vaccine group at low titers.”

- Line 352-354: "Together, these results show that Month 13 PRNT50 titer and Month 13 MN titer are consistent with the Prentice criteria for DENV-1 but not for the other serotypes." This is quite interesting. Why is this as supplemental figure (Fig. S2)?

Response: As mentioned above, we have moved Fig. S2 to the main text (Fig 6 in the revised manuscript). Moreover, we think this result is interesting enough that it is appropriate to include it in the abstract. We have added the following sentence to the abstract: 

“We also studied each assay as a valid surrogate endpoint based on the Prentice criteria, which supported each assay as a valid surrogate for DENV-1 but only partially valid for DENV-2, -3, and -4.” (lines 35-37)

- Line 360-361: "We previously showed that estimated VE against DENV-Any was approximately 25% for vaccine recipients with no seroresponse at Month 13…” Presumably, no seroresponse means titers <10? The model estimates for the undetectable titers (<10) are not shown in the figure, nor in the original figure S15 of Moodie et al. 2017. However, the <10 value is shown on the x-axis. The figures appear to only show titers from a value of 10, which is detectable. Is this just a plotting issue? On the microneut panel in Fig. 6, the x-axis only goes to 10, even though there are individuals with MN values <10.

Response: Thank you for the suggestion to clarify the definition of “no seroresponse” and to harmonize x-axis labels in Fig 6 (Fig 7 in the revised manuscript). Below we respond to the different elements of your comment:

We previously showed that estimated VE against DENV-Any was approximately 25% for vaccine recipients with no seroresponse at Month 13…” Presumably, no seroresponse means titers <10? 

Response: Yes. We have added the following text to the revised manuscript (lines 446-451):

“We previously showed that estimated VE against DENV-Any was approximately 25% for vaccine recipients with no seroresponse (defined as PRNT50 titer less than the assay lower limit of quantification, 10, for all four serotypes) at Month 13 and increased similarly with average Month 13 PRNT50 titer in baseline seronegative vs. baseline seropositive subgroups in CYD14 and CYD15 9–16-year-olds (baseline serostatus determined by the PRNT50 assay) (Fig 7A, 7B; reproduced with modification from [11]).”

The model estimates for the undetectable titers (<10) are not shown in the figure, nor in the original figure S15 of Moodie et al. 2017. However, the <10 value is shown on the x-axis. The figures appear to only show titers from a value of 10, which is detectable. Is this just a plotting issue? On the microneut panel in Fig. 6, the x-axis only goes to 10, even though there are individuals with MN values <10.

Response: Thank you for the suggestion to harmonize x-axis labels in Fig 6 (Fig 7 in the revised manuscript) across the VE curves by Month 13 average PRNT50 titer (Panels A, B) and the VE curves by Month 13 average MN titer (Panels C, D), all of which were generated using the Zhuang et al. method. The Zhuang et al. method was developed assuming an immune assay was measured with a lower limit of detection S=max(S*,c) (with c=10 for both the PRNT50 and MN assays) (i.e. the observed immune response is left-truncated at value 10), and on a model of disease risk conditional on the observed S. Thus, each curve shown in Fig 7 starts at x= 10, since that is the smallest value S could take.

Thus, even though panels A and B in Fig 6 of the originally submitted manuscript had a hash mark for “<10” on the leftmost part of the x-axis, there is not actually any data plotted with such a titer value. We see now that it could potentially be confusing to have this hashmark present, as there is no data there, and have removed the “<10” hash mark from panels A and B. As to the statement that “On the microneut panel in Fig. 6, the x-axis only goes to 10, even though there are individuals with MN values <10”, the method description above explains why each VE curve starts with 10, such that no VE value is reported for any MN value <10. If one adds a straight vertical line at a titer value of “10”, it is clear that there are no data points to the left of the line. We considered adding such a line to the curves but thought it would be visually distracting.

- Line 368-370: "At an average MN (PRNT50) Month 13 titer of 1000, VE estimates were 88% (85%) for baseline-seropositive vaccine recipients and 78% (76%) for baseline-seronegative vaccine recipients (Figure 6)." Why state the VE at this high a titer value? Based on the titer distributions, it was rare for individuals to have that high of titers in the trial even among controls. Perhaps it would be better to report VE based on the median titer observed.

Response: This is a good point. We now report VE in baseline-seropositive and baseline-seronegative vaccine recipients based on the median Month 13 PRNT50 titer. The added text is shown below: 

“Among CYD14 and CYD15 9–16-year-old vaccine recipients, VE estimates against DENV-Any at the median Month 13 average PRNT50 titer of 392 were 77% for the baseline-seropositive subgroup and 68% for the baseline-seronegative subgroup; at a Month 13 average MN titer of the same value (392), VE estimates were 82% for the baseline-seropositive subgroup and 70% for the baseline-seronegative subgroup (Fig 7). Thus, stratification by baseline serostatus of VE estimates by Month 13 titer in CYD14 and CYD15 9–16-year-old vaccine recipients yields relatively similar results for baseline-seropositive vs. baseline-seronegative subgroups, regardless of which assay is used to measure Month 13 titer.” (lines 456-463) 

- Part B Table S6: the model shows some very strong significant OR >>>1. (e.g. 52, 79, 830). What does this mean? Some of the effects seem to be for interaction terms? It is very difficult to interpret what this means without a description of what the terms are.

Response: Thank you for the suggestion to provide descriptions of what the terms are; we agree such descriptions would be helpful to the reader. We have provided the following footnotes to S6 Table (S5 Table in the revision):

aEstimates for interaction terms (indicated by notation X:Y) in the Odds Ratio column are ratios of odds ratios.

bAge.12.16 is the indicator of 12-16 years old compared to the reference category 2-5 years old.

cM13.PRNT.S1 is Month 13 PRNT50 DENV-1 titer, with similar notation for each neutralization assay and serotype.

dSero2.rate is the fraction of placebo group VCD endpoints that are of serotype 2, with similar notation for the other serotypes.

eM13.MN.Ave is Month 13 MN average titer to the 4 serotypes.

Discussion:

- Paragraph, 470: “It is unclear why nAb titer readouts did not perform equally well across serotypes as CoRs for their matched-serotype VCD endpoints.” Is this paragraph referring to Table 1? I thought the non-significant effects were DENV3 and DENV4? This paragraph mentions DENV1 and DENV3?

Response: We have substantially revised this paragraph, as detailed below (underlined text):

“While in CYD15 all titer readouts (serotype-specific and average) were significant CoRs for their corresponding VCD endpoints (matched-serotype VCD and DENV-Any VCD, respectively), it is unclear why some of the serotype-specific nAb titer readouts were stronger CoRs for their respective matched-serotype VCD endpoints than other serotype-specific nAb titer readouts were for their respective matched-serotype VCD endpoints. For example, for both assays, DENV-1 and DENV-3 titers tended to be less strong CoRs (albeit still significant CoRs) for serotype-matched VCD than DENV-2 and DENV-4 titers [PRNT50 – DENV-1: OR per log10 increment (95% CI) 0.31 (0.23, 0.42); DENV-3: 0.41 (0.27, 0.64) vs DENV-2: 0.18 (0.12, 0.27); 0.22 (0.09, 0.51). MN – DENV-1: 0.23 (0.15, 0.33); DENV-3: 0.23 (0.13, 0.41) vs DENV-2: 0.14 (0.08, 0.26); DENV-4: 0.08 (0.03, 0.19)] (Table 1). Differences between in vitro systems for assessing antibody-mediated DENV neutralization, which use cultured cell lines and laboratory DENV strains, versus neutralization of circulating DENV viral variants in the human body, may be relevant. For instance, the PRNT50 and MN assays assess neutralization of only one DENV strain per serotype. If participants are exposed to circulating viral variants that are neutralized less well (or better) than the assayed strain, the obtained titer for that serotype will be less representative of how well nAbs in that participant’s serum neutralize exposing viral variants. While it is generally assumed that nAb binding epitopes are conserved within serotypes, there is evidence supporting significant variation in neutralization across genotypes of a given serotype, particularly for DENV-1 and DENV-3 [38-40]. We speculate that a scenario in which contemporaneously circulating DENV-1 and DENV-3 strains are neutralized less well (or better) than the DENV-1 PUO-359 (isolated in 1980 in Thailand) and DENV-3 PaH881/88 (isolated in 1988 in Thailand) strains used in the PRNT50 and MN assays (i.e. the parental DENVs of the respective recombinant vaccine viruses) could explain why DENV-1 and DENV-3 titers tended to be weaker (yet still significant) CoRs versus DENV-2 and DENV-4 titers. 

The proportion of mismatched amino acid residues between the vaccine DENV inserts and the DENV sequences isolated from placebo group cases provides an assessment of the degree of match between circulating viral variants at the time of the trial and the vaccine strains, and may also be relevant to explain potential differences across serotypes in the strength of serotype-specific CoRs of matched-serotype VCD. We have previously analyzed these proportions by serotype and shown that, in CYD15, DENV-1 circulating strains were farthest from the DENV-1 vaccine insert, followed by DENV-2 circulating strains to the DENV-2 insert, DENV-3 circulating strains to the DENV-3 insert, and then DENV-4 circulating strains found to be closest to the DENV-4 vaccine insert [41]. The latter finding is consistent with DENV-4 titer being the strongest serotype-specific CoR of matched-serotype VCD in CYD15 (Table 1); however, these findings do not fully explain why DENV-2 titer tended to be a stronger CoR of DENV-2 VCD than DENV-1 titer was of DENV-1 VCD in CYD15. ” (lines 572-608)

- Line 517: "We next consider why individual-level classification accuracy using the different variable input." Is this referring to the super-learner algorithm?

Response: Yes. We have edited the sentence as follows (added text is underlined):

“We next consider why individual-level classification accuracy, using the best model identified by Super-Learner for each of four different covariate groups, was relatively limited.” (lines 643-644)

Minor comments:

- Line 60: "Estimated VE against these two endpoints was negative in baseline-seronegative individuals”. By negative, you mean had a negative vaccine efficacy, meaning worse off? This is unclear as written.

Response: Yes, by “negative estimated VE”, we mean “negative vaccine efficacy”. We have added the following text to the revision (underlined):

“Subsequent analyses of VE by baseline dengue serostatus showed high estimated VE against hospitalized VCD and against severe VCD over 60 months in baseline-seropositive individuals; however, estimated VE against these two endpoints was negative in baseline-seronegative individuals (i.e., vaccinated baseline-seronegative individuals were at higher risk of these two endpoints compared to unvaccinated baseline-seronegative individuals) [9].” (lines 66-68)

- Line 156: Please write out CV-AUC (cross validated area under the curve) the first time it appears.

Response: We have done this.

- "Lack of Month 13 sero-response at Month 13 approached zero for both assays (≤ 2%) for DENV-3 and DENV-4 (P>0.05 for both), but significant discordance remained between the MN and PRNT50 results for DENV-1 and DENV-2 (P<0.01 for both) (S1 Fig, panel C)." This sentence is confusing as written.

Response: We agree this sentence could be written more clearly and have modified the text as follows: 

“This pattern continued for the placebo group when Month 13 titers were assayed, i.e. higher percentages of participants tested Month 13-seronegative by the MN assay compared to the PRNT50 assay, across all five titer measurements (all P<0.01; S1 Fig, panel B). Similar results were obtained in the vaccine group for Month 13 DENV-1 and DENV-2 titers in that significantly greater percentages of participants tested Month 13-seronegative for DENV-1 and for DENV-2 by the MN assay compared to the PRNT50 assay (P<0.01 for both; S1 Fig, panel C). However, no significant difference in Month 13-seronegativity rates between the two assays was seen in the vaccine group for average titer, DENV-3, or DENV-4 (P>0.05 for all), with seronegativity rates ≤ 2% across the two assays (S1 Fig, panel C).” (lines 248-257)

- Line 232-234: "As one log10 increase in PRNT50 titer approximately equaled one log10 increase in MN titer (Figure 1), we used OR for comparing the two nAb readouts as CoRs.” Why not directly quantify this relationship? Why say approximately one log10 increase here?

Response: Thank you for pointing out that this sentence is potentially confusing as written. As the information in the sentence is not vital to understanding or interpreting the results, we have opted to delete this sentence from the revised manuscript.

- Fig. S2: the cumulative endpoint rate for Fig S2 C and D are quite different (DENV2): 0.05 for the PRNT vs. 0.014 for MN assay? Other y-axes of the PRNT50 vs. MN panels in this figure are more consistent.

Response: We first note that Fig S2 has been moved to the main text and is now Fig 6 in the revised manuscript. As we mentioned in the reviewer’s second comment under “Results” (i.e. beginning with “Line 351-352: “the DENV-2 and DENV-4 correlates of risk…”, there were very small numbers of DENV-2 cases among vaccine-recipients with no Month 13 DENV-2 PRNT50 seroresponse (0 DENV-2 cases among the 87 vaccine-recipients with no Month 13 PRNT50 DENV-2 seroresponse in CYD14 and 5 DENV-2 cases among the 264 vaccine-recipients with no Month 13 PRNT50 DENV-2 seroresponse in CYD15, with “no Month 13 PRNT50 DENV-2 seroresponse” defined as Month 13 PRNT50 DENV-2 titer < 10). Likewise, there were very small numbers of DENV-2 cases among vaccine-recipients with no Month 13 DENV-2 MN seroresponse (3 DENV-2 cases among the 739 vaccine-recipients with no Month 13 MN DENV-2 seroresponse in CYD14 and 34 DENV-2 cases among the 2774 vaccine-recipients with no Month 13 MN DENV-2 seroresponse in CYD15, with “no Month 13 MN DENV-2 seroresponse” defined as Month 13 MN DENV-2 titer < 10). These small numbers of DENV-2 cases imply that the estimated risks in the left tail of the curves shown in Fig 6, panels C and D, have wide confidence intervals. 

- Fig. 7: no color is shown in the legend for for D2 model panel A. I assume it should be blue?

Response: Thank you for pointing out this inadvertent oversight. Yes, the color should be blue. This has been added to the legend in panel A of Fig 7 (Fig 8 in the revised version of the manuscript). 

 

References

1. Rossenkhan R, Rolland M, Labuschagne JPL, et al.; Combining Viral Genetics and Statistical Modeling to Improve HIV-1 Time-of-infection Estimation towards Enhanced Vaccine Efficacy Assessment. Viruses 2019;11(7). doi: 10.3390/v11070607.

2. Magaret CA, Benkeser DC, Williamson BD, et al.; Prediction of VRC01 neutralization sensitivity by HIV-1 gp160 sequence features. PLoS Comput Biol 2019;15(4):e1006952. doi: 10.1371/journal.pcbi.1006952.

3. Gilbert PB, Luedtke AR; Statistical Learning Methods to Determine Immune Correlates of Herpes Zoster in Vaccine Efficacy Trials. J Infect Dis 2018;218(suppl_2):S99-S101. doi: 10.1093/infdis/jiy421.

4. Neidich SD, Fong Y, Li SS, et al.; Antibody Fc effector functions and IgG3 associate with decreased HIV-1 risk. J Clin Invest 2019;129(11):4838-4849. doi: 10.1172/JCI126391.

5. Sridhar S, Luedtke A, Langevin E, et al.; Effect of Dengue Serostatus on Dengue Vaccine Safety and Efficacy. N Engl J Med 2018;379(4):327-340. doi: 10.1056/NEJMoa1800820.

6. Price BL, Gilbert PB, van der Laan MJ; Estimation of the optimal surrogate based on a randomized trial. Biometrics 2018;74(4):1271-1281. doi: 10.1111/biom.12879.

7. Frangakis CE, Rubin DB; Principal stratification in causal inference. Biometrics 2002;58(1):21-9. doi: 10.1111/j.0006-341x.2002.00021.x.

8. Moodie Z, Juraska M, Huang Y, et al.; Neutralizing Antibody Correlates Analysis of Tetravalent Dengue Vaccine Efficacy Trials in Asia and Latin America. J Infect Dis 2018;217(5):742-753. doi: 10.1093/infdis/jix609.

---

## [Decision Letter · Decision Letter 1]

22 May 2020

Microneutralization Assay Titer Correlates Analysis in Two Phase 3 Trials of the CYD-TDV Tetravalent Dengue Vaccine in Asia and Latin America

PONE-D-19-30338R1

Dear Dr. Gilbert,

We are pleased to inform you that your manuscript has been judged scientifically suitable for publication and will be formally accepted for publication once it complies with all outstanding technical requirements.

With kind regards,

Ray Borrow, Ph.D., FRCPath

Academic Editor

PLOS ONE

Additional Editor Comments (optional):

Reviewers' comments:

Reviewer's Responses to Questions

**Comments to the Author**

1. If the authors have adequately addressed your comments raised in a previous round of review and you feel that this manuscript is now acceptable for publication, you may indicate that here to bypass the “Comments to the Author” section, enter your conflict of interest statement in the “Confidential to Editor” section, and submit your "Accept" recommendation.

Reviewer #1: All comments have been addressed

Reviewer #3: All comments have been addressed

Reviewer #4: All comments have been addressed

2. Is the manuscript technically sound, and do the data support the conclusions?

Reviewer #1: Yes

Reviewer #3: Yes

Reviewer #4: Yes

3. Has the statistical analysis been performed appropriately and rigorously? 

Reviewer #1: Yes

Reviewer #3: Yes

Reviewer #4: Yes

4. Have the authors made all data underlying the findings in their manuscript fully available?

Reviewer #1: Yes

Reviewer #3: No

Reviewer #4: Yes

5. Is the manuscript presented in an intelligible fashion and written in standard English?

Reviewer #1: Yes

Reviewer #3: Yes

Reviewer #4: Yes

6. Review Comments to the Author

Reviewer #1: The authors have responded to the questions raised by reviewers in a thoughtful and methodical manner. I have no further concerns.

Reviewer #3: The authors addressed my previous set of statistical questions and clarifications with satisfaction; I have no further comments.

Reviewer #4: The authors have been comprehensive in responding to my review of their manuscript. I have no further comments.

7. PLOS authors have the option to publish the peer review history of their article (what does this mean?). If published, this will include your full peer review and any attached files.

Reviewer #1: Yes: Anuja Mathew

Reviewer #3: No

Reviewer #4: No

---

## [Editor Report · Acceptance letter]

2 Jun 2020

PONE-D-19-30338R1 

Microneutralization Assay Titer Correlates Analysis in Two Phase 3 Trials of the CYD-TDV Tetravalent Dengue Vaccine in Asia and Latin America 

Dear Dr. Gilbert:

I'm pleased to inform you that your manuscript has been deemed suitable for publication in PLOS ONE. Congratulations! Your manuscript is now with our production department. 

Kind regards, 

on behalf of

Prof. Ray Borrow 

Academic Editor

PLOS ONE